# Structure of Rap1b bound to talin reveals a pathway for triggering integrin activation

Liang Zhu[1,2], Jun Yang[1], Thomas Bromberger[3], Ashley Holly[1], Fan Lu[1,2], Huan Liu[1], Kevin Sun[1], Sarah Klapproth[3], Jamila Hirbawi[1], Tatiana V. Byzova[1], Edward F. Plow[1], Markus Moser[3] & Jun Qin[1,2]

Activation of transmembrane receptor integrin by talin is essential for inducing cell adhesion. However, the pathway that recruits talin to the membrane, which critically controls talin's action, remains elusive. Membrane-anchored mammalian small GTPase Rap1 is known to bind talin-F0 domain but the binding was shown to be weak and thus hardly studied. Here we show structurally that talin-F0 binds to human Rap1b like canonical Rap1 effectors despite little sequence homology, and disruption of the binding strongly impairs integrin activation, cell adhesion, and cell spreading. Furthermore, while being weak in conventional binary binding conditions, the Rap1b/talin interaction becomes strong upon attachment of activated Rap1b to vesicular membranes that mimic the agonist-induced microenvironment. These data identify a crucial Rap1-mediated membrane-targeting mechanism for talin to activate integrin. They further broadly caution the analyses of weak protein–protein interactions that may be pivotal for function but neglected in the absence of specific cellular microenvironments.

[1] Department of Molecular Cardiology, Lerner Research Institute, Cleveland Clinic, 9500 Euclid Avenue, Cleveland, OH 44195, USA. [2] Department of Biochemistry, Case Western Reserve University, Cleveland, OH 44106, USA. [3] Max-Planck-Institute of Biochemistry, Department of Molecular Medicine, 82152 Martinsried, Germany. Liang Zhu, Jun Yang and Thomas Bromberger contributed equally to this work. Correspondence and requests for materials should be addressed to M.M. (email: moser@biochem.mpg.de) or to J.Q. (email: qinj@ccf.org)

The adhesion of cells to the extracellular matrix (ECM) is essential for regulating a variety of physiological or pathological responses such as platelet aggregation, blood clotting, stroke, wound-healing, and cancer metastasis. Such adhesion critically depends upon integrins, a class of cell surface receptors that are (α/β) heterodimers with each subunit containing a small cytoplasmic tail (CT), a transmembrane segment, and a large extracellular domain. In unstimulated cells, integrin adopts an inactive conformation with low affinity for ECM ligands but, upon agonist stimulation, integrin CT is bound by intracellular proteins, in particular talin and/or kindlin, which trigger global conformational change of the extracellular domain to acquire high affinity for ECM ligand and initiate firm adhesion. This process, widely regarded as inside-out signaling or integrin activation, has been the central topic of cell adhesion research for nearly three decades[1–6]. Extensive efforts have been focused on investigating the mechanism of integrin activation by talin—a major cytoskeletal protein comprising an N-terminal head domain (talin-H) and a C-terminal rod domain (talin-R). Talin-H contains four subdomains F0, F1, F2, and F3 in which F1–F3 constitutes a so-called FERM (4.1/ezrin/radixin/moesin)-like domain. Talin-R is made up of 13 consecutive helical bundles followed by a C-terminal dimerization domain (Fig. 1). Intact talin adopts an inactive conformation where the major integrin-binding site located in talin-H F3 domain is masked by talin-R (autoinhibition)[7–10]. Talin may be unmasked via multiple mechanisms to bind integrin β CT[7, 8, 11–13] and trigger the receptor activation[14–17].

While a great deal has been learned about talin, a fundamental issue still remains unresolved: how is cytoplasmic talin recruited to plasma membrane—a crucial step for talin to bind and activate integrin? A previous dogma, largely derived from studying prototypic integrin α_{IIb}β_3 from platelets, is that agonist stimulation activates small GTPase Rap1 on the membrane to recruit an effector called RIAM that in turn recruits talin[5, 18]. However, while both Rap1 and talin are highly abundant[19], RIAM is present at very low levels in platelets with no other homologs detected[20]. More importantly, mice lacking RIAM are viable and the platelet functions including the α_{IIb}β_3 activation are totally normal[20–22]. By contrast, Rap1 and talin are both essential for integrin activation[4, 5, 18]. For example, ablation of the Rap1 isoform Rap1b severely impairs α_{IIb}β_3 activation and platelet aggregation[23]—defects that were also observed in talin knockout mice[24, 25]. These

observations strongly suggest a RIAM-independent engagement between Rap1 and talin to regulate integrin activation. Interestingly, previous studies reported a direct interaction between Rap1 and talin in mammals[26] and in Dictyostelium[27], but the affinity of the interaction was found to be low especially for that in mammals ($K_d \sim 0.14$ mM)[26]. Since weak protein–protein interactions (PPIs) are typically assumed to be nonspecific and RIAM was widely regarded as a key bridge between Rap1 and talin, the Rap1/talin interaction was hardly studied at the mechanistic level. Through comprehensive structural, biochemical, and cell biological analyses, we discovered that talin is actually a distinct effector of Rap1 that recruits talin to membrane and triggers integrin activation. We further found that, while being weak in conventional binary binding assays, Rap1/talin interaction became strong when activated Rap1 was anchored to membrane that mimics the cellular condition. Our results thus uncover a mechanism of membrane-targeting of talin by Rap1 and also highlight the importance of weak PPIs that may be neglected in the absence of specific cellular microenvironments.

## Results

**Rap1 recognizes talin-F0 domain in a GTP-dependent manner.** Rap1 is known to contain two highly homologous isoforms called Rap1a and Rap1b[28–31]. In this study, we choose to focus on examining the interaction between the platelet-rich Rap1b and talin-F0. Like other small GTPases, Rap1b exists in inactive (GDP-bound) and active (GTP-bound) states. We first used a highly sensitive NMR technique called heteronuclear single quantum coherence (HSQC)[32] to examine the Rap1b/talin-F0 interaction. Figure 2a and Supplementary Fig. 1a show that Rap1b loaded with GTP analog GMP-PNP induced residue-specific chemical shift changes of [15]N-talin-F0. Inactive Rap1b loaded with GDP also binds to talin-F0 but at weaker affinity as evidenced by smaller chemical shift changes than those with GMP-PNP (Fig. 2b). Thus, talin-F0 binds favorably to the active form of Rap1b but, as shown in Fig. 2c, the binding affinity is quite low ($K_d \sim 162$ μM) as previously concluded[26]. On the other hand, talin-F0 does not bind H-Ras (Supplementary Fig. 1b) that is near 90% homologous in terms of binding interface (overall similarity is ~76%) to Rap1b (see also Supplementary Fig. 1c), suggesting that despite being weak, the talin-F0/Rap1b interaction is specific. To further investigate the specificity of Rap1b/talin-F0

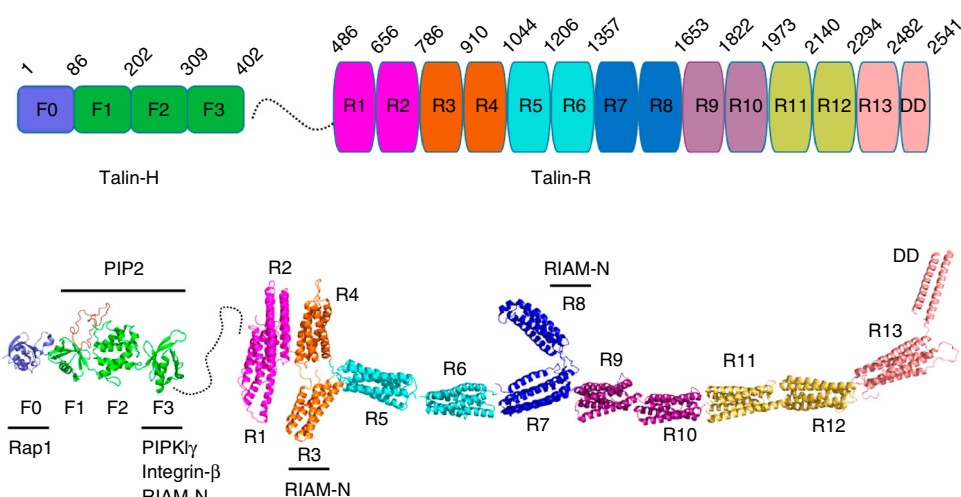

**Fig. 1** Domain organization and a linear structural model of talin. The model was generated from known crystal and NMR structures of talin fragments and shown in cartoon representation. Critical talin-binding partners involved in talin membrane recruitment and activation are indicated in the figure. *PIP2* phosphatidylinositol-4,5-bisphosphate, *RIAM-N* the N terminus of RIAM, *PIPKIγ* type I phosphatidylinositol phosphate kinase isoform-γ

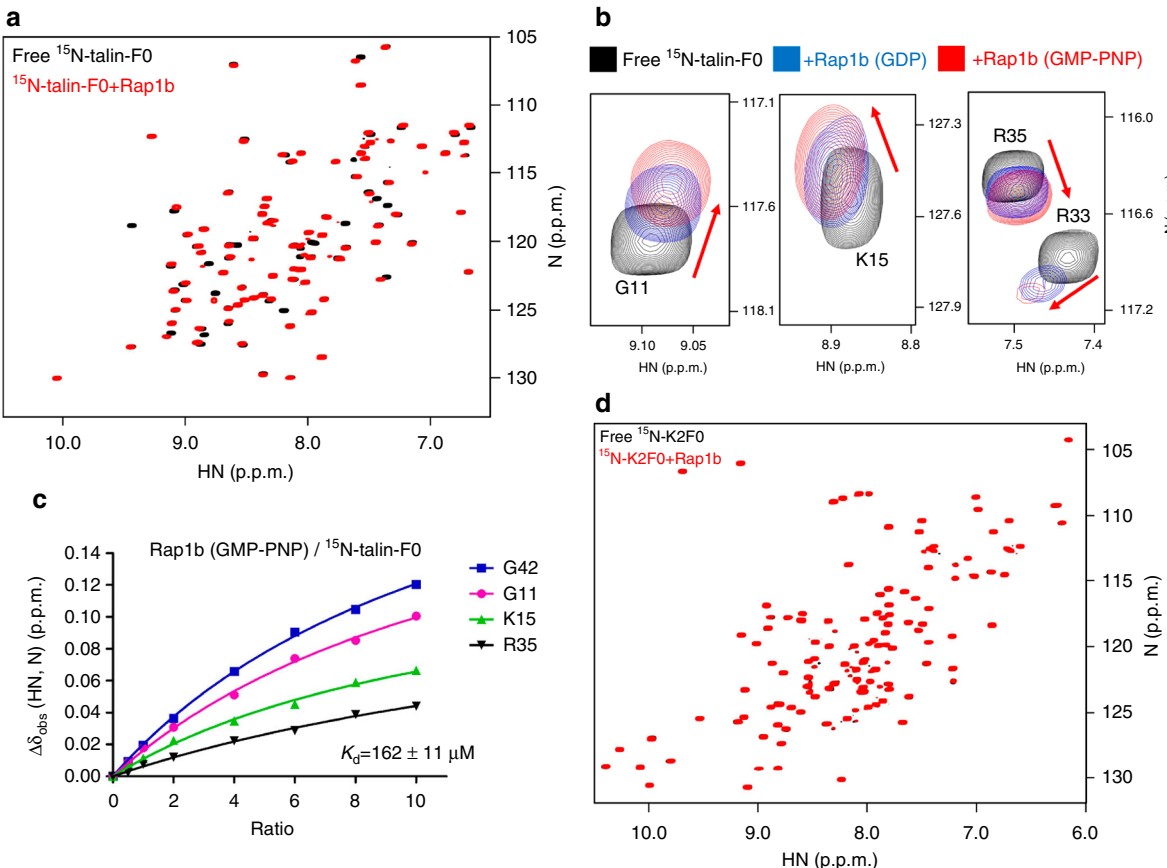

**Fig. 2** Rap1b/talin-F0 interaction is GTP-dependent and specific but is of weak affinity. **a** The HSQC spectra of 50 μM $^{15}$N-labeled talin-F0 in the absence (black) and presence of 125 μM GMP-PNP loaded Rap1b (red). **b** The HSQC spectra (four representative residues were shown) of 50 μM $^{15}$N-labeled talin-F0 in the absence (black) and presence of 125 μM GDP loaded Rap1b (blue) or GMP-PNP loaded Rap1b (red). Note that Rap1b-GDP induced the same overall pattern of chemical shift changes of $^{15}$N-labeled talin-F0 as Rap1b-GMP-PNP does but with less peak shifts and broadenings, indicating a weaker affinity. **c** The affinity of Rap1b (GMP-PNP)/talin-F0 interaction measured by HSQC titration. **d** The HSQC spectra of 50 μM $^{15}$N-labeled kindlin2-F0 in the absence (black) and presence of 125 μM GMP-PNP loaded Rap1b (red)

interaction, we examined the Rap1b binding to F0 domain of another integrin activator kindlin2 that is homologous to talin-H (Supplementary Fig. 2a)[33], but no interaction was detected (Fig. 2d). Since talin-H contains FERM domain (F1–F3) prior to F0 and F1–F2 of KRIT1 FERM domain was shown to bind Rap1b[34], we also wondered whether F1–F2 of talin FERM would be involved in binding to Rap1b. However, Supplementary Fig. 2b shows that Rap1b does not interact with talin FERM F1–F2. Overall, these data suggest that Rap1b engages with talin-F0 in a highly specific manner. Interestingly, NMR (Fig. 3a, Supplementary Figs. 1a and 2c) and Glutathione S-transferase (GST) pull-down (Fig. 3b) experiments revealed that active Rap1b also binds talin-H and full-length talin in the same manner as to talin-F0. Given that full-length talin is autoinhibited with F0 being unmasked[7], these data suggest that active Rap1b specifically recognizes the F0 domain of resting talin.

**Rap1b/talin interface exhibits distinct features**. To understand how Rap1b recognizes talin, we decided to pursue the total structure of active human Rap1b (~20 kDa) in complex with talin-F0 (~10 kDa). Solution NMR is particularly suitable for structure determination of weak protein complexes[32]. By performing a series of three-dimensional (3D) heteronuclear experiments using a combination of $^{1}$H/$^{15}$N/$^{13}$C-labeled and $^{2}$H/$^{15}$N-labeled samples, we were able to obtain nearly complete resonance assignment, 3422 intramolecular NOEs, and 78 intermolecular NOEs (Supplementary Fig. 3), which allowed

determination of the complex structure (Fig. 4a and Supplementary Table 1). Figure 4b displays the cartoon diagram of the overall complex in which individual subunits, Rap1b and talin-F0, adopt a Ras family GTPase fold and an ubiquitin-like fold, respectively, as expected. The ubiquitin-like fold of talin-F0 has similarity to Ras-association (RA) domain present in known Rap1 effectors such as KRIT1, RIAM, and c-Raf1 (Fig. 4c). However, despite this similar fold, talin-F0 has little sequence homology with these Rap1b RAs (Fig. 4d) with only three positively charged residues being relatively conserved (highlighted in red in Fig. 4d). The detailed Rap1b/talin-F0 interface is summarized in Fig. 4e and Supplementary Fig. 4a and compared with other Rap1b effectors (Supplementary Table 2). Majority of the contacts in the Rap1b/talin-F0 complex are different from other Rap1b/effector complexes (Supplementary Table 2) apparently due to the scant sequence homology of talin-F0 with other Rap1b effectors (Fig. 4d). A particularly distinct feature in the Rap1b/talin-F0 complex is the existence of a hydrophobic core between I36/P37 of talin-F0 and V21/I27/V29 of Rap1b (Fig. 4e and Supplementary Fig. 4b). Such core was not observed in other Rap1b/effector complexes. To understand how Rap1b but not the highly homologous H-Ras binds talin-F0 (Supplementary Fig. 1b), we mutated I27 and K31 of Rap1b, which closely contact with talin-F0, into corresponding H-Ras residues, Histidine and Glutamate, respectively. Both mutations showed reduced binding to talin-F0 (Supplementary Fig. 4c). By contrast, both Rap1b and H-Ras bind equally well to RIAM as shown before[35] as also confirmed in

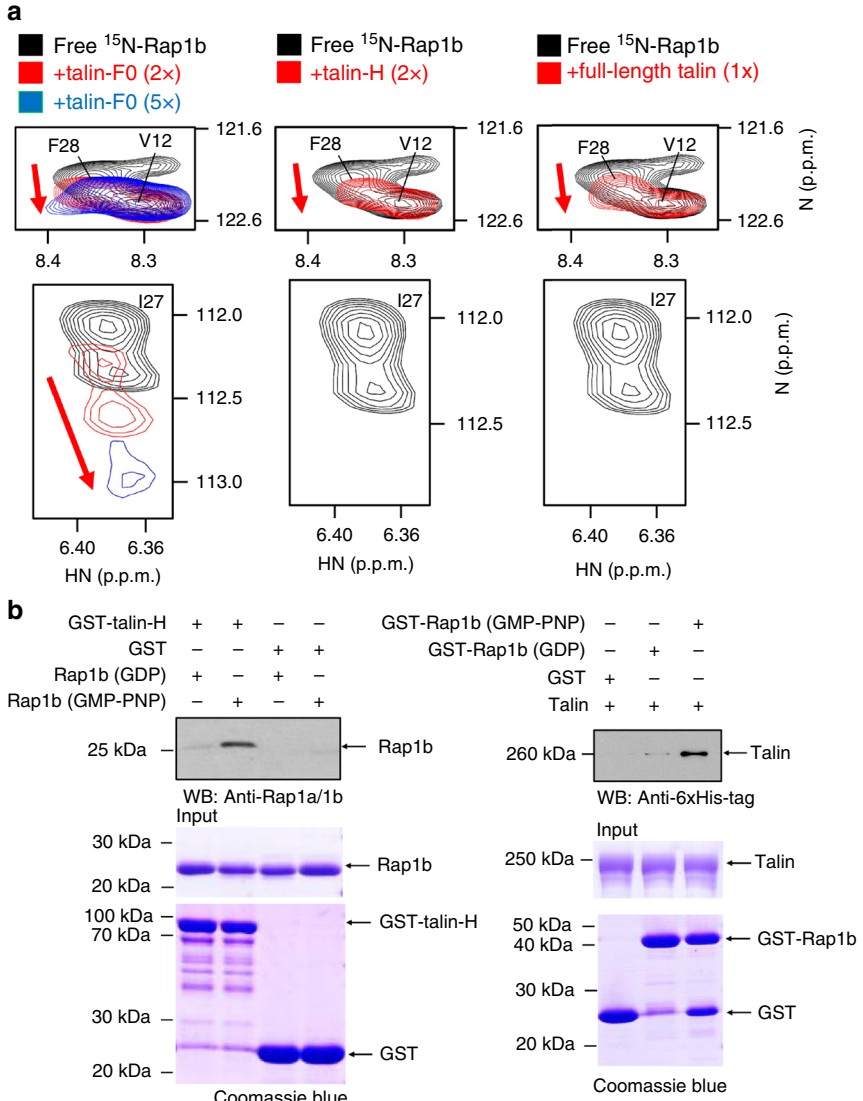

**Fig. 3** Rap1b recognizes F0 domain of talin in a GTP-dependent manner. **a** The HSQC spectra (representative regions were shown) of 45 μM GMP-PNP loaded $^{15}$N-labeled Rap1b (1–167) in the absence (black) and presence of 90 μM talin-F0 (red), 225 μM talin-F0 (blue), 90 μM talin-H (red), or 45 μM full-length talin (red). Note that the peak of I27 was broadened in the presence of talin-H or full-length talin. **b** GST pull-down assays to show that Rap1b interacts with both talin-H and full-length talin in a GTP-dependent manner. Full blot/gel images are shown in Supplementary Figs. 11 and 12

Supplementary Fig. 4d and Supplementary Fig. 16. These data provide insight into the high specificity of the talin binding to Rap1b despite the weaker affinity than that between talin and RIAM (Supplementary Fig. 4e).

Further analysis revealed that, while the Rap1b/talin-F0 interface exhibits distinct features (Fig. 4e, Supplementary Fig. 4a, and Supplementary Table 2), the overall topology of the complex shares similarity with other Rap1b/effector complexes (Fig. 4b, c). A key determinant for this similarity appears to be the antiparallel β-sheet between Rap1b β2 strand and talin-F0 β2 strand (Fig. 4b and Supplementary Fig. 4a), as also present in other Rap1b/effector complexes (Fig. 4c). In addition, charge–charge interactions in the talin-F0/Rap1b complex, which involve talin-F0 K7 with Rap1b E37 and talin-F0 K15/R35 with Rap1b D33, are relatively conserved in other Rap1b/effector complexes (Fig. 4d and Supplementary Table 2). By contrast, kindlin2-F0, which is not a Rap1b effector (Fig. 2d) but is highly homologous to talin-F0 (Supplementary Fig. 2a), lacks two positively charged residues in the corresponding positions (Fig. 4d).

In summary, the Rap1b/talin-F0 interface is quite unique yet resembles other Rap1b/effector complexes in overall topology. These structural and binding data suggest that talin is a direct effector of Rap1. Interestingly, the interface residues are highly conserved in both talin/Rap1 isoforms and in different species (Supplementary Fig. 5a, b), suggesting that Rap1/talin recognition is evolutionarily conserved.

**Rap1b binding to talin is crucial for integrin activation.** To further evaluate the importance of the Rap1b/talin interaction, we performed the structure-based analysis and mutated two critical interface residues K15 and R35 of talin-F0 (Fig. 4e and Supplementary Fig. 4a) into alanines. Figure 5a and Supplementary Fig. 6 show that K15A/R35A double mutations (DM) drastically reduced the Rap1b interaction to talin-F0 as well as talin-H. Since talin-H is the unit responsible for binding and activating integrin[5,36], we next tested the impact of the Rap1b-binding defective mutations on its ability to activate $\alpha_{IIb}\beta_3$—the prototypic integrin widely used to study integrin activation. To this end, we

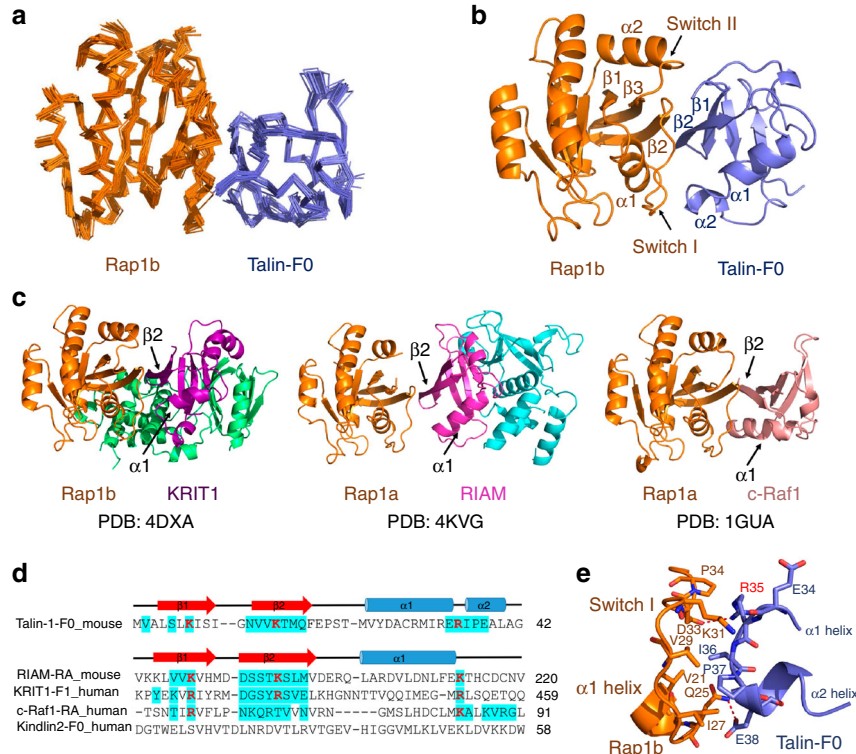

**Fig. 4** Solution structure of Rap1b/talin-F0 complex by NMR. **a** Superposition of 20 calculated Rap1b/talin-F0 complex structures with lowest energies (shown in ribbon representation). **b** Cartoon representation of the Rap1b/talin-F0 complex structure with the lowest energy. **c** Current solved complex structures of Rap1 and its effector proteins (shown in cartoon representation). **d** Structure-based sequence alignment of talin-F0 and RA domain containing Rap1b effector proteins or kindlin2-F0 (only binding interfaces were shown). Residues involved in the binding interface with a cutoff of 4 Å are highlighted in cyan. Conserved residues are colored in red. **e** Detailed interaction diagram between the α1 helix and switch I of Rap1b and the α2 helix of talin-F0. Hydrogen bond or salt bridge is represented by red dashed line

transfected wild-type (WT) talin-H or talin-H DM into Chinese hamster ovary (CHO) A5 cells stably expressing $\alpha_{IIb}\beta_3$ (Supplementary Fig. 7a). Figure 5b shows that, as compared to the WT talin-H, the talin-H mutant-induced $\alpha_{IIb}\beta_3$ activation was substantially reduced. These results provide strong evidence that the interaction of talin-H with Rap1b is crucial for the talin-H-mediated integrin activation. They also explain a previously reported phenomenon where deletion of F0 domain in talin-H significantly impaired the integrin activation[37]. To further demonstrate that the impaired interaction between talin DM and Rap1 has an impact on integrin-mediated functions, we retrovirally transduced talin-null fibroblasts (talin[1/2dko]), which show a strong defect in cell adhesion and spreading[38], with C terminally ypet-tagged talin WT as well as talin DM. The transduced cells were FACS-sorted for equal talin and talin DM expression as confirmed by flow cytometry and western blotting (Supplementary Figs. 7b and 17). Re-expression of both WT talin and talin DM induced cell spreading of talin[1/2dko] cells with WT and mutant talin localizing to paxillin-positive focal adhesions (FAs; Supplementary Figs. 7c, d). We then performed adhesion assays on various integrin ligands and observed that talin DM only partially rescued cell adhesion of talin[1/2dko] cells to fibronectin (FN), laminin-111 (LN), and vitronectin (VN) compared to WT talin, whereas integrin-independent adhesion to poly-L-lysine was unchanged (Fig. 5c). As expected, talin[1/2dko] cells expressing ypet as a control hardly adhered to any integrin ligand (Fig. 5c). Importantly, expression of mutant talin did not alter the expression levels of surface integrins nor those of RIAM and Rap1 (Supplementary Figs. 7b, e). Next, we plated the cells on FN and measured cell spreading for 4 h. While ypet-transduced talin[1/2dko] cells remained roundish and did not spread, talin DM cells

displayed a significant spreading defect compared to talin WT-transduced cells (Fig. 5d).

We hypothesize that the reduced PAC1 binding, cell adhesion, and spreading of talin DM-expressing cells are due to impaired integrin activation caused by deficient talin recruitment to the integrin site at the plasma membrane. To test this hypothesis experimentally, we seeded WT talin and talin DM-expressing cells on FN-coated micropatterns for 4 h and measured the number and size of paxillin-positive FAs. This analysis reveals that indeed the FA area and number per cell are significantly reduced in cells expressing talin DM compared to those expressing WT talin (Figs. 5e–g). The mean FA size (Fig. 5h) was also reduced by talin DM, although with a $p$ value of 0.09. To further test whether impaired talin/Rap1 interaction affects the recruitment of talin DM to FAs, we measured the intensity of ypet fluorescence within FAs and correlated it to the total cellular ypet fluorescence intensity. We observed a significantly reduced relative intensity of ypet signal within FAs in talin DM-transduced cells, which suggests that the talin/Rap1b interaction is important for talin recruitment to FAs (Fig. 5i). Collectively, our data provide strong evidence for the physiological importance of direct Rap1/talin interaction in integrin regulation.

**Membrane-anchored Rap1b shows enhanced binding to talin.** While the forgoing data demonstrated the unique specificity of Rap1b/talin interaction and its importance in regulating integrin-mediated functions, a key issue still remains: how can such interaction with ~0.16 mM affinity be effective since protein concentration in cells is typically much lower than 0.16 mM? We attempted to address this issue from two angles. First, both Rap1

and talin are highly abundant in cells. For example, recent proteomic studies showed that Rap1b and talin belong to the top most abundant proteins in platelets with copy numbers of ~200,000, which are nearly 1/4 of the most abundant platelet protein actin[19]. The high levels of Rap1 and talin in platelets are consistent with our protein expression analysis (Fig. 6a). Given that the average concentration of actin is above 200 μM in

platelets[39], the estimated concentration of Rap1b or talin in platelets should be at least 50 μM. By contrast, the RIAM level is very low in both human and murine platelets (Fig. 6a), which was also confirmed by quantitative mass spectrometry with a copy number of only 162[19]. Thus, the high concentrations of Rap1 and talin increase the chance for their direct interaction to occur. Second, given that Ras family GTPases including Rap1 are all

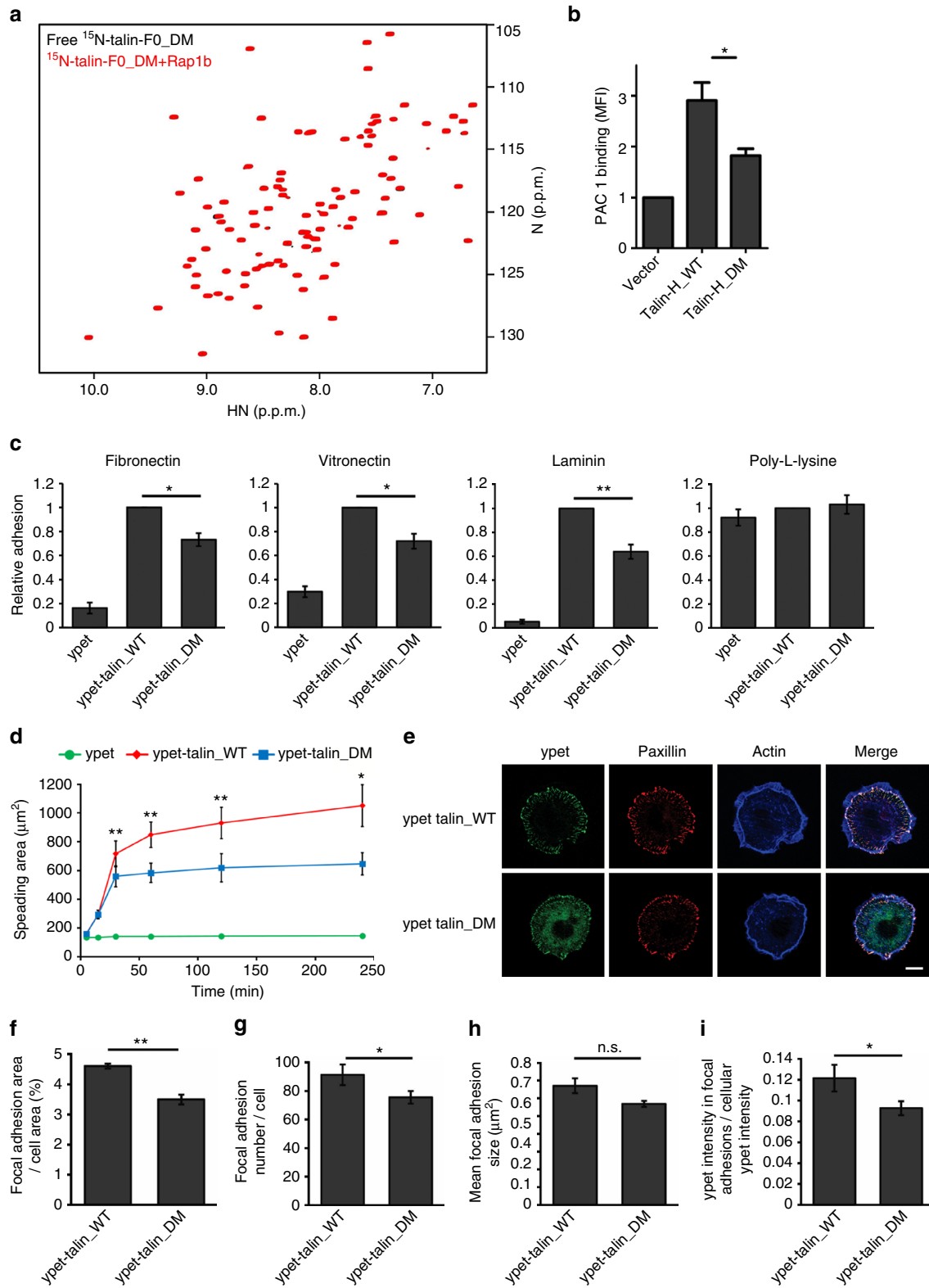

known to be attached to membrane via Cys prenylation of their C-terminal CAAX motif[29] and talin-H also binds to membrane via a large positively charged surface on F1, F2, and F3 domains[8, 40], we reasoned that Rap1 and talin-H might be more tightly associated if Rap1 were prenylated with membrane to mimic the true cellular microenvironment (Fig. 6b). Our Rap1b/talin-F0 structure superimposed with the structure of talin-H supports this possibility in that the CAAX containing C terminus of Rap1b and the positively charged surface of talin-H align well to all face toward the membrane surface (Fig. 6b). To address this possibility experimentally, we prepared membrane vesicles with anchored Rap1b using the well-established thiol-maleimide crosslinking method[41] (Supplementary Fig. 8a). As shown in Fig. 6c, membrane-anchored Rap1b indeed bound much more robustly to talin-H than Rap1b unanchored to membrane. Isothermal titration calorimetry (ITC) experiments revealed a $K_d$ of ~1.52 μM for membrane-anchored active Rap1b to bind to talin-H (Supplementary Fig. 8b), which indicates a more than two orders of magnitude stronger binding than that measured without membrane (Fig. 2c and Supplementary Fig. 4e). Consistently, the membrane-binding capacity of talin-H was also substantially enhanced by Rap1 anchored to the membrane, but such enhancement was diminished for talin-H DM that is defective for binding to Rap1 (Fig. 6d). Importantly, the binding of membrane-anchored Rap1b to talin-H is GTP-dependent as shown by two independent methods: GST pull-down assay (Supplementary Figs. 8c and 18) and co-sedimentation assay (Supplementary Figs. 9a and 19). Such GTP dependence indicates that the effective membrane targeting of talin requires active Rap1b. These findings provide key evidence for talin being a bona fide Rap1 effector and suggest how agonist-activated and membrane-anchored Rap1 can effectively recruit talin to the membrane surface. Although full-length talin binds to membrane more weakly than talin-H possibly due to some degree of auto-inhibition for the former, membrane with anchored Rap1b clearly binds more potently to full-length talin contrasting to the scenario where Rap1b was not attached to the membrane (Fig. 6e). Similar to membrane-mediated Rap1b/talin-H binding (Supplementary Fig. 9a), membrane-anchored Rap1b also interacts with full-length talin in a GTP-dependent manner (Supplementary Fig. 9b). Importantly, since full-length talin adopts an auto-inhibited conformation[7, 10], which may be unmasked through a pull–push mechanism by binding to the membrane[8], our data further suggest that Rap1 not only promotes the membrane targeting but also leads to activation of talin via this mechanism (Fig. 6b). In order to test this hypothesis, we performed NMR-based competition experiments by examining the interaction of $^{15}N/^2H$-labeled inhibitory talin rod domain 9 (talin-R9) with talin-H in the absence and presence of membrane-anchored Rap1b. Supplementary Fig. 9c shows that, while talin-H induces residue-specific chemical shift changes of talin-R9 as the signature of the autoinhibition[8], such chemical shift changes were suppressed by membrane-anchored Rap1b, thus demonstrating the talin-unmasking process.

## Discussion

Rap1 was discovered nearly three decades ago[42], and its two isoforms Rap1a and Rap1b (with ~95% identity, see also Supplementary Fig. 5b) are regarded as the key membrane-associated small GTPases to target proteins to the plasma membrane to regulate diverse cellular responses[28–31]. A large body of data have indicated that Rap1 is critically involved in activation of integrins[5, 18, 43], but the immediate downstream effector of Rap1 has been highly elusive. Our studies have now obtained an important clue on this longstanding mystery. We found that the activated Rap1b interacts with the F0 domain of talin—the key mediator of integrin activation, and we determined the NMR structure of the Rap1b/talin-F0 complex. Despite little sequence homology between talin-F0 and other Rap1 effectors, the overall topology of the complex was found to resemble those known Rap1/effector complexes. Moreover, disruption of the Rap1b/talin interface substantially impaired the integrin $\alpha_{IIb}\beta_3$ activation, cell adhesion, and cell spreading. Interestingly, we found that, while being weak measured in conventional binary conditions[26], the Rap1b/talin interaction became quite strong when Rap1b was anchored to membrane—a condition mimicking the cellular microenvironment of Ras family GTPases[41] and also favoring talin unmasking as indicated previously[8]. Overall, our data suggest a crucial pathway by which agonist-activated Rap1 directly promotes the membrane recruitment of talin, which further leads to the talin unmasking and integrin activation (Fig. 6b). Our data also provide an explanation of why Rap1 and talin are essential for integrin activation, whereas RIAM is dispensable for activation of most integrins expressed on many cell types as indicated by the recent series of genetic studies[20–22]. On the other hand, RIAM, which is highly abundant in leukocytes, was shown to play an important role in regulating leukocyte integrin function[21, 22] by possibly forming the so-called "MIT complex" with talin and integrin[44], indicating that the direct Rap1/talin pathway plays a minor role in regulating the leukocyte $\beta_2$ integrins. The fact that talin DM still partially supports cell adhesion and spreading of talin$^{1/2dko}$ cells (Fig. 5c, d) further indicates the existence of other compensatory pathways including "Rap1-RIAM-talin" and "PIPKIγ/PIP2-talin"[7, 8, 11]. More detailed investigations are needed to understand how these different pathways are turned on or cooperated in a spatiotemporal manner. Nevertheless, our data, combined with the genetic evidence of Rap1 and talin in integrin activation, suggest that direct Rap1–talin interaction critically regulates integrin activity.

In addition to uncovering the physiological role of the Rap1b/talin interaction, our findings also bear broad implications in the analyses of PPIs. Tens of thousands of PPIs in cells form a web of intricate communication networks for regulating diverse

**Fig. 5** The Rap1b binding to talin is crucial for integrin activation. **a** The HSQC spectra of 50 μM $^{15}N$-labeled talin-F0_DM (K15A, R35A) in the absence (black) and presence of 125 μM GMP-PNP loaded Rap1b (red). **b** Integrin activation assay in CHO A5 cells, which stably express integrin $\alpha_{IIb}\beta_3$. Double mutations (K15A, R35A) in talin-H substantially decreases integrin activation. The data are shown as means ± S.E.M. from four independent experiments. *$p < 0.05$. **c** Static adhesion of talin$^{1/2dko}$ fibroblasts, expressing either ypet alone, ypet-tagged talin WT, or ypet-tagged talin DM (K15A, R35A). Number of adherent cells was quantified by measuring absorbance of crystal violet staining. Values measured for talin WT-transduced cells were set to one in each of six independent experiments. Values are given as mean ± S.E.M. *$p < 0.05$ and **$p < 0.01$. **d** Spreading of talin$^{1/2dko}$ fibroblasts expressing ypet (green), ypet-tagged talin WT (red), or ypet-tagged talin DM (blue) on a fibronectin-coated surface, measured 5, 15, 30 min, 1, 2 and 4 h after plating. $N = 5$. **e** Confocal images of ypet-tagged talin WT or ypet-tagged talin DM cells on FN-coated round micropatterns stained for paxillin (red) and F-actin (blue). Ypet signal is shown in green. Scale bar 10 μm. Focal adhesion area per cell area (**f**), focal adhesion number per cell (**g**), and focal adhesion size (**h**) of ypet-tagged talin WT and ypet-tagged talin DM cells on FN-coated micropatterns. **i** Ypet fluorescence intensity within paxillin-positive focal adhesion area in relation to the cellular ypet fluorescence. $N = 5$; 10–15 talin WT and talin DM-expressing cells in each measurement were analyzed. All values are given as mean ± S.E.M. *$p < 0.05$ and **$p < 0.01$

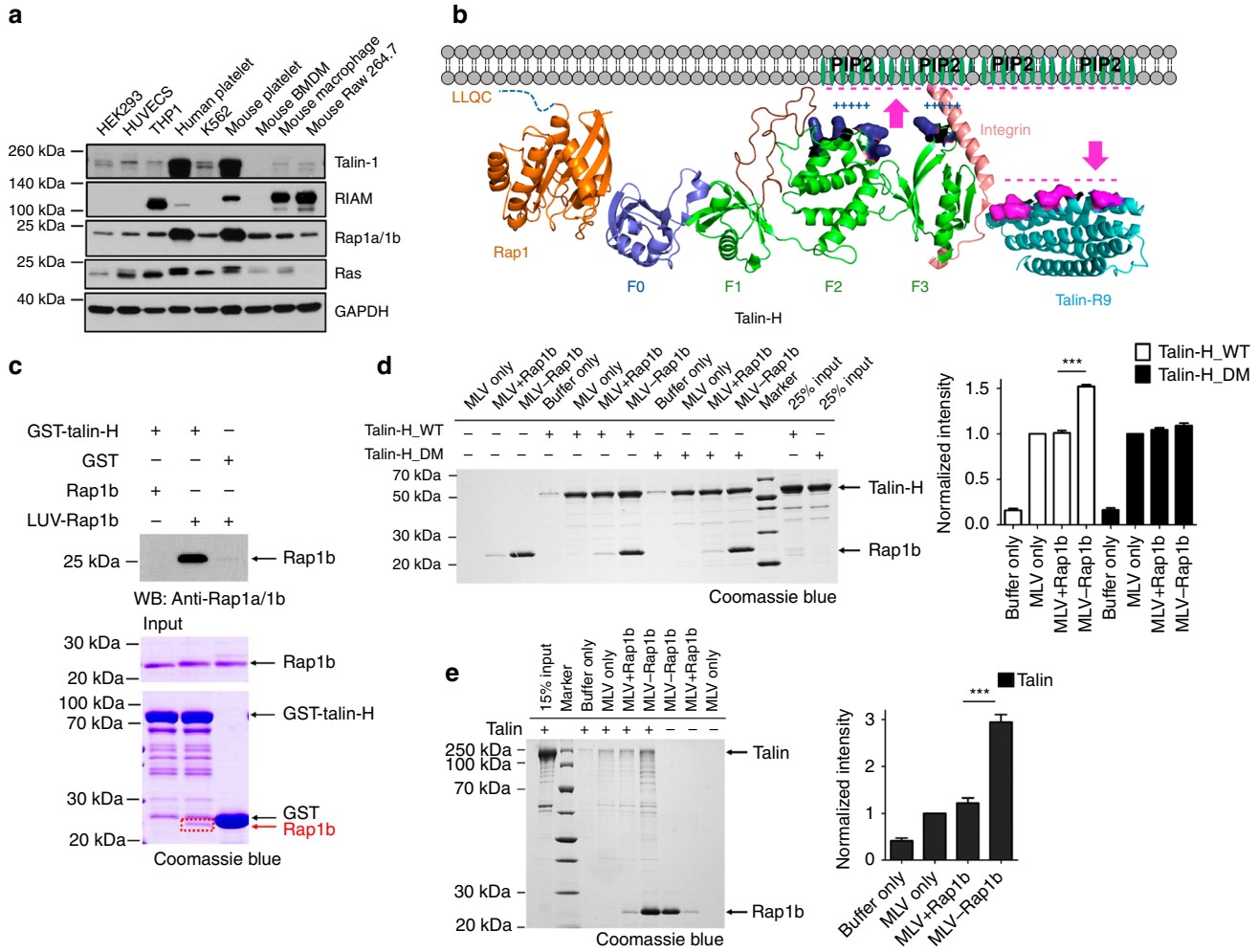

**Fig. 6** Mechanism of talin recruitment by Rap1. **a** Protein expression levels of talin, RIAM, Rap1, Ras GTPases, and GAPDH in various types of cell lines indicated in the figure. Full blots are shown in Supplementary Fig. 13. **b** A model of membrane-associated Rap1b recruiting talin for integrin activation. Proteins are shown in cartoon representation. Talin F1 loop is colored in maroon, the positively charged residues of talin-F2F3 domains (green), which interact with PIP2, are colored in deep blue and shown in surface representation, and the negatively charged residues of talin-R9 (cyan) are colored in magenta and shown in surface representation. **c** GST pull-down assay to show the robustly enhanced interaction between membrane-anchored Rap1b and talin-H. Rap1b was loaded with GMP-PNP in this assay and "LUV-Rap1b" represents that Rap1b was anchored to large unilamellar vesicles (LUVs). LUV-Rap1b could be easily pulled down by GST-talin-H and the band was visible in Coomassie blue staining gel (boxed in red) and intense in WB analysis, while free Rap1b pulled down by GST-talin-H was hardly seen in the same condition. Full blot/gel images are shown in Supplementary Fig. 14. **d** Left, a representative vesicle co-sedimentation assay showing that the interaction between talin-H_WT and membrane is enhanced by around 1.5-folds when Rap1b (GMP-PNP) is attached to membrane but not for talin-H_DM. "MLV + Rap1b" represents that Rap1b remained as free form in solution without being attached to multilamellar vesicles (MLVs), which were pre-incubated with β-mercaptoethanol. "MLV-Rap1b" represents that Rap1b was anchored to MLVs. Right, the quantification of four independent experiments. The intensity of talin-H_WT or talin-H_DM band was normalized to that of corresponding "MLV only" group and the data are shown in means ± S.E.M. ***$p < 0.0001$. Full gel image is shown in Supplementary Fig. 15. **e** Left, a representative vesicle co-sedimentation assay showing that the interaction between full-length talin and membrane is significantly enhanced by around 3-folds when Rap1b (GMP-PNP) is attached to membrane. Right, the quantification of four independent experiments. The intensity of talin band was normalized to that of the "MLV only" group and the data are shown as means ± S.E.M. ***$p < 0.0001$. Full gel image is shown in Supplementary Fig. 15

biological processes[45]. Interestingly, recent proteomic analyses have suggested that weak PPIs dominate in these cellular PPI networks[46]. However, because cellular protein concentrations, which are typically estimated on whole-cell volumes, are low (e.g., nM), weak binary PPIs, as detected by conventional in vitro methods, were commonly treated to be nonspecific and therefore disregarded for further investigations. Nevertheless, these weak PPIs may be enhanced and they exert crucial functions at specific microenvironments, which is strongly demonstrated in our study. Specifically, while isolated Rap1b and talin-F0 binds at sub-mM affinity, the binding became very strong in the presence of membrane that mimics the cellular microenvironment. As

illustrated in Fig. 6b, the membrane-anchoring ability of both Rap1b and talin may increase the local concentration of both proteins, thereby markedly strengthening their association. Other types of cellular strategies may also exist to enhance the association of otherwise weak binary interactions, which remain to be further investigated. With the completion of human genome and rapidly growing data of weak PPIs[46], our study emphasizes cautious treatment of a wide variety of weak PPIs by including specific microenvironments. In this regard, NMR is particularly valuable for deciphering the detailed structural basis of weak PPIs[32] in vitro, laying down the foundation for further analysis of the interaction in certain networks both in vitro and in vivo, as

exemplified in our study. Such approach would complement with other techniques such as crystallography and CryoEM that are advantageous for elucidating large high-affinity complexes, together, facilitating a thorough and unbiased view of how proteins function in cells.

## Methods

**Plasmid constructs and mutagenesis.** The following constructs for bacterial expression were used in this study: mouse talin-F0 (1–86) subcloned into a pHis-1 vector, mouse full-length talin subcloned into a pET28t vector, human kindlin2-F0 (1–105) subcloned into a pGST-1 vector, mouse talin-H (1–429) subcloned into a pET28t, and a pGST-1 vector, mouse RIAM-RAPH (149–438) subcloned into a pGST-1 vector, human full-length Rap1b subcloned into a pGST-1 vector, and C-terminal truncated Rap1b (1–167) subcloned into a pET28t vector. Human wild-type Rap1b and H-Ras (G12V) subcloned into pET28a vectors (6xHis tag) were kind gifts from Dr. Matthias Buck at Case Western Reserve University. Construct mutagenesis was conducted by using QuickChange lightning site-directed mutagenesis Kit (Agilent Technologies). Mutated constructs generated in this study include talin-F0_DM (K15A, R35A), talin-H_DM (K15A, R35A), Rap1b (G12V), Rap1b (1–167, G12V), Rap1b (G12V, I27H), Rap1b (G12V, K31E), Rap1b_TM1 (C51S, C118S, C141S), and Rap1b_TM2 (G12V, C51S, C118S, C141S). Note that pET28t is a modified pET28a vector where the thrombin cleavage site is substituted with a TEV protease cleavage site. Primers used in this study are listed in Supplementary Table 3.

**Protein expression and purification.** Recombinant proteins with fusion tag (6xHis tag or GST tag) were expressed in *Escherichia coli* BL21 (DE3) strain (New England Biolabs). Typically, bacteria was initially grown in 50 ml Lysogeny broth (LB) medium and then amplified in 1 ~ 2 l LB at 37 °C. The culture was induced by 0.4 mM Isopropyl β-D-1-thiogalactopyranoside (IPTG) at 20 °C or room temperature (RT) for overnight when it reached an $A_{600}$ of 0.7. The pellet was then collected and suspended in buffer and frozen at −80 °C. For protein purification, the pellet was lysed by incubation with lysozyme and sonication. After high-speed centrifugation, the supernatant was subjected to affinity column purification by using either nickel or GST gravity column. Ion exchange chromatography was an optional purification step depending on protein purity (in this study, full-length Rap1b was purified by anion exchange method using Hi-trap Q column). Gel filtration was always performed in the final step by using either Superdex-75 (16/60) or Superdex-200 (16/60; GE Healthcare). Note that Superose-6 (10/300) was used for full-length protein purification. Purified protein was checked by SDS-PAGE. Proteins with 90% purity or above were used in this study. According to experimental requirement, fusion tag (either His-tag or GST tag) was removed by TEV protease during protein purification if necessary. Isotope-labeled ($^{15}N$ and/or $^{13}C$) proteins were achieved by employing minimal medium with $^{15}NH_4Cl$ and/or $^{13}C$-glucose as the sole nitrogen and carbon sources. $^{2}H$-labeled protein was achieved by using the minimal medium prepared in 99.8% $D_2O$ with $^{2}H$-glucose as the carbon source. Protein concentration was measured by absorbance at 280 nm or Pierce 660 nm protein assay reagent (Thermo Fisher Scientific).

**GTPase nucleotide exchange.** Non-hydrolyzable analogs of GDP and GTP, named Guanosine 5′-[β-thio] diphosphate trilithium salt (GDP-β-S) and Guanosine 5′-[β, γ-imido] triphosphate trisodium salt (GMP-PNP), respectively, were purchased from Sigma-Aldrich. In this study, wild-type Rap1b loaded with GDP-β-S was used as an inactive form, and Rap1b (G12V) or H-Ras (G12V) loaded with GMP-PNP was used as an active form. Purified wild-type Rap1b (50 μM), Rap1b (G12V) or H-Ras (G12V) was first incubated with 2 mM EDTA and 2 mM GDP-β-S or GMP-PNP at RT for 35 min, MgCl2 was then added in to reach a final concentration of 7 mM and incubated for another 30 min. After that, they were kept on ice and concentrated into a high concentration for experimental use.

**NMR 2D-HSQC and HSQC titration.** 2D-HSQC experiments were performed on Bruker 600 MHz NMR spectrometer, and samples containing 50 μM $^{15}N$-labeled protein were studied. For $^{15}N$-labeled talin-F0 or kindlin2-F0, experiments were performed at 25 °C in buffer containing 20 mM NaH2PO4/Na2HPO4 (pH 6.6), 50 mM NaCl, 5 mM MgCl2, 2 mM dithiothreitol (DTT), and 5% D2O. For $^{15}N$-labeled Rap1b (1–167), experiments were performed at 25 °C in buffer containing 25 mM NaH2PO4/Na2HPO4 (pH 6.8), 150 mM NaCl, 5 mM MgCl2, 2 mM DTT, and 5% D2O. Chemical shift change ($\Delta\delta_{obs\ [HN,N]}$) was calculated with the equation $\Delta\delta_{obs\ [HN,N]} = [(\Delta\delta_{HN}W_{HN})^2 + (\Delta\delta_N W_N)^2)]^{1/2}$, where $W_{HN}$ and $W_N$ are weighting factors based on the gyromagnetic ratios of $^1H$ and $^{15}N$ ($W_{HN} = 1$ and $W_N = 0.154$) and $\Delta\delta$ (p.p.m.) = $\delta_{bound} - \delta_{free}$. For HSQC titration, samples containing 45 μM $^{15}N$-labeled talin-F0 with increasing amount of GMP-PNP-loaded Rap1b (G12V) were used to collect 2D-HSQC spectra. Dissociation constant ($K_d$) was achieved by fitting the chemical shift changes into the equation as shown below[47]. Four independent well-resolved residues were selected for fitting and an average $K_d$ was obtained.

$$\Delta\delta_{obs} = \Delta\delta_{max}\{(K_d + [P]_t + [L]_t) - [(K_d + [P]_t + [L]_t)^2 - 4 \times [P]_t \times [L]_t]^{1/2}\} / 2[P]_t$$
($\Delta\delta_{max}$, maximum chemical shift change; $[P]_t$, labeled protein concentration; $[L]_t$, ligand or titrant concentration).

**GST pull-down.** A volume of ~ 25 μg purified GST or GST-fused protein was immobilized on 15 μl glutathione-Sepharose 4B resin (beads) via incubation on a rotator for 1.5 h at 4 °C in the binding buffer containing 25 mM Tris-HCl (pH 7.5), 150 mM NaCl, 5 mM MgCl2, 1 mM DTT, 0.01% Nondiet P-40, and supplemented with Complete EDTA-free Protease Inhibitor (Roche, Indianapolis, IN). Desired amount of prey protein was then added in and incubated with beads for another 2 h at 4 °C. After that, the beads were washed by 600 μl binding buffer twice and subjected to be denatured by adding 35 μl 2 × SDS loading buffer and boiling for 5 min. After centrifuging down, supernatants were resolved by SDS-PAGE or western blotting. All the pull-down experiments were performed in triplicate independently.

**Western blotting and antibodies.** After SDS-PAGE, samples were transferred onto a 0.45 μm polyvinylidene fluoride (PVDF) membrane (Millipore, Billerica, MA). The membrane was blocked with 5% non-fat milk in TBST buffer overnight at 4 °C. After that, the membrane was incubated with primary antibody at RT for 3 h and then incubated with horeseradish peroxidase (HRP)-conjugated secondary antibody at RT for 1 h. The blots were detected by Pierce ECL Western Blotting Substrate (Thermo Fisher Scientific). The following primary antibodies were used in this study: 6xHis epitope tag antibody (Catalog# MA1-21315, Thermo Fisher Scientific), Rap1a/Rap1b (26B4) Rabbit mAb (Catalog# 2399, Cell Signaling Technology), Ras (27H5) Rabbit mAb (Catalog# 3339, Cell Signaling Technology), GAPDH (D16H11) Rabbit mAb (Catalog# 5174, Cell Signaling Technology), Anti-Talin-1 antibody (Catalog# ab57758, Abcam Inc.), and Anti-RIAM antibody (EPR2806; Catalog# ab92537, Abcam Inc.). The following secondary antibodies were used in this study: anti-mouse IgG, HRP-linked antibody (Catalog# 7076, Cell Signaling Technology), and anti-rabbit IgG, HRP-linked antibody (Catalog# 7074, Cell Signaling Technology). Primary antibodies were used at 1:1000 dilution, and secondary antibodies were used at 1:3000 dilution.

**Assignment of Rap1b and talin-F0 and their bound forms.** C-terminal truncated Rap1b (1–167) with G12V mutation was used in all the following structure determination studies. For assignment of Rap1b, $^{15}N$/$^{13}C$-labeled Rap1b was purified and the N-terminal 6xHis tag was removed during purification. Standard triple-resonance experiments were conducted with 0.6 mM GMP-PNP-loaded Rap1b. Rap1b sequential assignment was performed using PASA software[48]. For assignment of talin-F0, the complete chemical shift–assignment table of talin-F0 from Biological Magnetic Resonance Data Bank (accession code 15458) was downloaded and used in this study[26]. Amide proton to nitrogen to α-carbon correlation (HNCA) experiment was performed on 1.0 mM $^{15}N$/$^{13}C$-labeled talin-F0 to confirm the assigned resonances. All these experiments were performed on Bruker 600 MHz NMR spectrometer, and at 25 °C in the buffer that contained 20 mM NaH2PO4/Na2HPO4 (pH 6.6), 50 mM NaCl, 5 mM MgCl2, 2 mM d-DTT, and 5% D2O. For assignments of the bound Rap1b and talin-F0 forms, the chemical shifts of majority of residues were readily transferred from the free forms with slight adjustments as they are similar to those of the free forms. A few residues (e.g., I27, R41) exhibited duplex peaks in the free form of Rap1 likely due to local conformational flexibility, and most of these peaks were merged into single peaks in the complex. All assignments were also verified by through-bond NOESY experiments of the complex.

**3D-NOESY experiments.** The following samples were prepared: (1) 0.6 mM $^{15}N$/$^{13}C$-labeled Rap1b in the presence of 0.9 mM unlabeled talin-F0, (2) 0.5 mM $^{15}N$/$^{13}C$-labeled talin-F0 in the presence of 0.7 mM unlabeled Rap1b, (3) 0.5 mM $^{15}N$/100% $^{2}H$-labeled talin-F0 in the presence of 0.7 mM unlabeled Rap1b, and (4) 0.5 mM $^{15}N$/$^{13}C$-labeled talin-F0 in the presence of 0.7 mM unlabeled Rap1b prepared in 99.8% D2O. The following NOESY experiments were performed using the above samples: (i) 3D $^{15}N$/$^{13}C$-edited NOESY experiment (120 ms mixing time) with sample 1, (ii) 3D $^{15}N$/$^{13}C$-edited NOESY experiment (120 ms mixing time) with sample 2, (iii) 3D $^{15}N$-edited NOESY experiment (300 ms mixing time) with sample 3, and (iv) 3D $^{15}N$/$^{13}C$-filtered NOESY experiment (120 ms mixing time) with sample 4. All these experiments were performed on Bruker 850 or 900 MHz spectrometers at the same condition as mentioned above.

**Structure determination.** All NMR data were processed and analyzed using nmrPipe[49], PIPP[50], and Sparky[51]. Distance restraints (NOE constraints) were obtained from 3D-NOESY experiments mentioned above, and Xplor-NIH[52] was used to calculate the complex structure. In brief, the individual structures of Rap1b and talin-F0 in complex form were calculated first based on the NOEs collected from sample 1 (2308 NOEs) and sample 2 (1114 NOEs), respectively. 302 dihedral-angle restraints for Rap1b derived from TALOS[53] were applied during Rap1b structure calculation. The initial templates for Rap1b and talin-F0 were derived from PDB database with PDB codes of 4DXA and 3IVF, respectively. A total of 78 unambiguous intermolecular NOEs collected from sample 3 and sample 4 were used to calculate complex structure (also see Supplementary Table 1 for structural

statistics). After refinements, a total of 50 final structures were calculated and the 20 lowest-energy structures were selected for analysis. The quality of structure was evaluated with PROCHECK[54]. Structures are visualized by PyMOL 1.3 (Schrödinger LLC.).

**Membrane-anchored Rap1b preparation.** To prepare large unilamellar vesicles (LUVs), 1-palmitoyl-2-oleoyl-sn-glycero-3-phosphocholine (POPC), L-α-phosphatidylinositol-4,5-bisphosphate (PI(4,5)P$_2$), and 1,2-dipalmitoyl-sn-glycero-3-phosphoethanolamine-N-[4-(p-maleimidomethyl)cyclohexane-carboxamide] (16:0 PE MCC) were purchased from Avanti Polar Lipids. LUVs, which consist of 96% POPC, 1% PIP$_2$, and 3% PE MCC, were prepared by extrusion. In brief, lipids were first dissolved together in chloroform, and then the chloroform was removed under a stream of nitrogen followed by overnight vacuum pumping. The lipid film was suspended in buffer containing 25 mM NaH$_2$PO$_4$/Na$_2$HPO$_4$ (pH 6.8), 100 mM NaCl, and 5 mM MgCl$_2$, and subjected to homogenization with a few freeze–thaw cycles. LUVs were finally formed by extruding the lipid suspension ~20 times through two stacked 0.1 mm polycarbonate filters. Rap1b was tethered to LUVs via Michael addition between the thiol group of Cys 180 and the maleimide group of PE MCC[41, 55]. To mimic the physiological condition that only the C-terminal Cys 180 of Rap1b is attached to membrane and meanwhile to avoid nonspecific chemical interactions, all the cysteines of Rap1b except Cys 180 were mutated to serines. Rap1b_TM1 (C51S, C118S, and C141S) and Rap1b_TM2 (G12V, C51S, C118S, and C141S) were generated, purified, and loaded with GDP-β-S and GMP-PNP, respectively, as described above. Rap1b_TM2 was first confirmed to interact with [15]N-talin-F0 by HSQC (Supplementary Fig. 10a). To make Rap1b-anchored LUVs, Rap1b_TM1 or Rap1b_TM2 was incubated with freshly prepared LUVs in 1:6 molar ratio overnight at RT in the same buffer. The reaction was terminated by adding β-mercaptoethanol to a final concentration of 5 mM. The achievement of membrane-anchored Rap1b was confirmed by Native-PAGE combined with SDS-PAGE (Supplementary Figs. 10b and 20). Unanchored Rap1b was dialyzed out in the same buffer using a dialyzer with 100 kDa cutoff membrane (Harvard Apparatus) before use.

**Vesicle co-sedimentation.** Multilamellar vesicles (MLVs) rather than LUVs are used in this study since MLVs are larger in size and could be pelleted down easily. MLVs, which consist of 87% POPC, 10% PIP2, and 3% PE MCC were prepared similarly to LUVs as described above but without extrusion. MLVs with anchored Rap1b (either GDP-β-S or GMP-PNP) were also prepared similarly as described above. Talin-H (10 μM) or full-length talin was incubated with 10 μM "MLV-Rap1b (Rap1b-anchored MLVs)", "MLV only", and "MLV + Rap1b (MLVs with unanchored Rap1b)" separately in a "7 × 21 mm" Polycarbonate tube (Beckman Coulter Inc.) with a total volume of 30 μl at RT for 15 min. Samples were then centrifuged down at 22,000 r.p.m. for 35 min. Membrane pellet was dissolved in 2 × SDS loading buffer and subjected to SDS-PAGE analysis. Gels were stained with Coomassie blue, and then scanned and quantified with an Odyssey CLx Imaging System (LI-COR Inc.). The intensities of protein bands were quantified by the fluorescent signal detected from the 700 nm channel according to the manufacturer's protocol. Experiments were performed in quadruplets. Statistical significance was tested by two-tailed unpaired t-test.

**Isothermal titration calorimetry.** ITC experiments were performed using a Microcal iTC 200 system (GE Healthcare Life Sciences) at 25 °C. Before experiments, all samples were dialyzed into the same buffer that contained 25 mM NaH$_2$PO$_4$/Na$_2$HPO$_4$ (pH 6.8), 100 mM NaCl, and 5 mM MgCl$_2$ overnight at 4 °C. Talin-H (200 μM) in the syringe (~ 40 μl) was injected 20 times in 2.0 μl aliquots into the sample-cell (~220 μl) containing 15 μM LUV-Rap1b (GMP-PNP). As controls (Supplementary Fig. 10c), talin-H was titrated into LUV only, Rap1b only, or buffer only, and talin-H_DM was titrated into LUV-Rap1b (GMP-PNP) using the same condition. ITC Data were analyzed by fitting to a single-site binding model with Origin Software.

**Microscale thermophoresis analysis.** The affinity of the interaction between GMP-PNP-loaded GFP-Rap1b (G12V) and RIAM-RAPH, talin-F0 or talin-H was measured by using a Monolith NT.115 instrument (NanoTemper Technologies). Rap1b (G12V) was subcloned into a pRSET vector (Thermo Fisher Scientific), which expresses an N-terminal 6xHis tag and a C-terminal GFP fusion tag. RIAM-RAPH, talin-F0, or talin-H was prepared at different concentrations through 1:1 serial dilution and then mixed with 100 nM GFP-Rap1b (G12V) in buffer containing 50 mM Tris-HCl (pH 7.5), 100 mM NaCl, 5 mM MgCl$_2$, 2 mM DTT, and 0.05% Tween 20. The samples were incubated at RT for 10 min and then loaded into microscale thermophoresis (MST) standard or premium capillary tubes. The fluorescent signal of GFP fusion tag was detected by the blue channel of the instrument according to the manufacturer's protocol, and the data were collected using 20% LED power and 40% MST powder at RT. Three independent experiments were performed, and the collected MST data were used to fit the dissociation constant ($K_d$) by using the MO. Affinity Analysis software (NanoTemper Technologies).

**Integrin activation assay.** Wild-type talin-H or talin-H_DM (K15A, R35A) was subcloned into a pEGFP-C1 vector. CHO A5 cells stably expressing integrin α$_{IIb}$β$_3$ were transfected with empty vector, wild-type talin-H, or talin-H_DM using Lipofectamine 2000 reagent (Life Technologies). After 24 h, cells were stained with anti-α$_{IIb}$β$_3$ activation-specific mAb PAC1 (1:100 dilution; Catalog# 340535, BD Biosciences) at RT for 40 min, and later incubated with Alexa Flour 647 Goat Anti-Mouse IgM (Catalog# 115-607-020, Jackson Immunology Laboratories; 1:3000 dilution) on ice for 30 min. After wash, cells were fixed and subjected to flow cytometry analysis. Values for PAC1 binding were reported as median fluorescent intensities and the data were normalized to control with empty vector only and are presented by the means ± S.E.M. The two-tailed unpaired t-test was performed to calculate the p value using GraphPad software.

**Generation of cell lines and cell culture conditions.** Murine talin1 and 2 double-knockout fibroblasts (talin[1/2dko])[38] were retrovirally transduced with pLPCX expression construct containing YPET alone, C terminally YPET-tagged talin-1 cDNA (talin WT), or YPET-tagged double-mutant talin-1 (talin DM; K15A, R35A) using the Phoenix retroviral expression system. Phoenix cells were transfected using a conventional calcium phosphate protocol and five infection cycles were performed by transferring the phoenix cell culture supernatant supplemented with 5 μg ml$^{-1}$ polybrene to talin[1/2dko] cells[56, 57]. Cells were FACS-sorted using a FACSAria™ Iiu cell sorter (BD Biosciences, Heidelberg, Germany) to isolate cells with comparable expression levels. Cells were characterized by western blotting using mouse α-GFP (home-made), rabbit anti-talin-1(Catalog# sc-15336, Santa Cruz Biotechnology Inc.), rabbit anti-Rap1 (Catalog# sc-65, Santa Cruz Biotechnology Inc.), rabbit anti-RIAM (Catalog# ab92537, Abcam Inc.), mouse anti-GAPDH (Catalog# CB1001-500UG, Millipore), goat anti-rabbit-HRP (Catalog# 111-035-144, Jackson ImmunoResearch Laboratories), and goat anti-mouse-HRP (Catalog# 115-035-003, Jackson ImmunoResearch Laboratories) antibodies following standard protocols. Rabbit anti-talin-1, rabbit anti-Rap1, and rabbit anti-RIAM antibodies were used at a dilution of 1:1000. Mouse anti-GAPDH, goat anti-rabbit-HRP, and goat anti-mouse-HRP antibodies were used at a dilution of 1:20,000. Cells were cultured under standard conditions in DMEM GlutaMAX™ (Thermo Fisher Scientific) supplemented with 10% fetal bovine serum, 100 U ml$^{-1}$ penicillin, 100 μg ml$^{-1}$ streptomycin, and non-essential amino acids (all from Thermo Fisher Scientific).

**Adhesion and spreading assays.** For adhesion assays, polystyrol flat-bottom 96-well microplates (Greiner Bio-One, Frickenhausen, Germany) were coated with 10 μg ml$^{-1}$ laminin (Sigma) in HBSS (Thermo Fisher Scientific), 5 μg ml$^{-1}$ FN, 0.01 % poly-L-lysine (both Sigma), or 5 μg ml$^{-1}$ VN (STEMCELL Technologies, Köln, Germany) in coating buffer (20 mM Tris-HCl pH 9.0, 150 mM NaCl, 2 mM MgCl$_2$) overnight at 4 °C. After blocking with 3% bovine serum albumin in PBS for 30 min at room temperature, $4 \times 10^4$ cells in DMEM containing 0.1 % FBS, and 25 mM HEPES were seeded per well, incubated for 1 h, and washed with PBS. Adherent cells were fixed with 4% paraformaldehyde (PFA) for 15 min and stained with 5 mg ml$^{-1}$ crystal violet in 2% ethanol for 30 min. After washing, remaining crystal violet was dissolved in 2% SDS and quantified by measuring absorbance at 595 nm using a microplate reader (Tecan, Männedorf, Switzerland). All experiments were performed in quadruplets. For spreading analysis, cells were plated on FN (5 μg ml$^{-1}$)-coated dishes and phase contrast pictures were taken using an EVOS™ FL Auto life cell microscope (Thermo Fisher Scientific) 5, 15, 30, 60, 120, and 240 min after seeding the cells. Cell spreading area of 30 cells per group at each time point were measured using ImageJ software (US National Institutes of Health). Statistical significance was tested by two-tailed paired t-test.

**Flow cytometry.** Integrin surface expression was determined by FACS using a LSRFortessa™ X-20 flow cytometer (BD Biosciences). Staining and measurement were performed in PBS supplemented with 2% FBS and 2 mM EDTA. Cells were incubated with biotinylated anti-integrin antibodies and subsequently with Cy5-labeled strepdavidin (Catalog# 016-170-084, Jackson ImmunoResearch Laboratories). Data were analyzed using FlowJo software. The following antibodies were used for flow cytometry: hamster IgM anti-integrin β1 (Catalog# 13-0291-82, eBioscience), rat IgG2a anti-integrin α6 (Catalog# 13-0495-82, eBioscience), rat IgG1 isotype control (Thermo Fisher Scientific), hamster IgG isotype control (Thermo Fisher Scientific), hamster IgG anti-integrin β3 (Catalog# 553345, BD Pharmingen), rat IgG2a anti-integrin α5 (Catalog# 557446, BD Pharmingen™), rat IgG1 anti-integrin αV (Catalog #104104, BioLegend), hamster IgM isotype control (BioLegend), and rat IgG2a isotype control (BioLegend). All listed antibodies were biotinylated and used at 1:200 dilution.

**Immunostaining and FA analysis.** Cells were cultured on FN-coated glass coverslips (Thermo Fisher Scientific) overnight or on FN-coated micropatterns (disc-shaped with 1100 μm$^2$ area; Cytoo, Grenoble, France) for 5 h and fixed for 10 min with 4% PFA. Mouse anti-paxillin with a dilution of 1:300 (Catalog# 610051, BD Transduction Laboratories, Heidelberg, Germany), goat anti-mouse-Cy3 with a dilution of 1:400 (Catalog# 115-165-146, Jackson ImmunoResearch Laboratories), and Phalloidin-Alexa Fluor 647 with a dilution of 1:100 (Catalog# A22287, Thermo Fisher Scientific) were used for immunostainings. Images were acquired using a

Leica TCS SP5 X confocal microscope (Leica Microsystems, Wetzlar, Germany) equipped with ×63 numerical aperture (NA) 1.40 oil objective lenses and Leica Confocal Software (LAS AF). All pictures were processed with Photoshop (Adobe Systems, San José, CA, USA). FA area was defined by paxillin staining and quantified using ImageJ software. For quantitative analysis of talin recruitment to FAs, the total ypet signal intensity within the FA area was normalized to the total ypet fluorescence of the whole cell. Statistical significance was tested by two-tailed paired $t$-test.

**Cell lysate preparations**. Various types of cells indicated in Fig. 6a were isolated from blood or cultured cells. Cells were washed once with cold PBS, and lysed by adding buffer that contained 50 mM Tris-HCl (pH 6.8) and 1% SDS. The cell lysates were boiled for 5 min, and the concentration of total protein in each cell lysate was quantified with Pierce BCA Protein Assay Kit (Thermo Fisher Scientific). Approximately 8 µg total protein for each lysate was loaded onto SDS-PAGE gels for western blotting analysis.

**Data availability**. Accession codes: The complete chemical shift–assignment tables of free Rap1b, bound Rap1b, and bound talin-F0 have been deposited in the Biological Magnetic Resonance Bank (BMRB) with the accession code 30353. The coordinates of Rap1b/talin-F0 complex structure have been deposited in the Protein Data Bank (PDB) with the code 6BA6. Other data are available from the corresponding authors upon reasonable request.

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

## Acknowledgements

We thank Reinhard Fässler for providing us with talin1/2 double-knockout fibroblasts. This work was supported by NIH grants R01HL58758, R01GM62823 to J.Q., P01HL073311 to E.F.P., the Max-Planck-Society, and the SFB1123 TP A08 to M.M. We thank Jianmin Liu, Koichi Fukuda, Sujay Ithychanda, and Xi-An Mao for assistance and useful discussions.

## Author contributions

L.Z., M.M. and J.Q. conceived this study. L.Z. performed all biochemical/biophysical studies including NMR 2D-HSQC experiments, GST pull-down assays, ITC experiments, and MST assay. L.Z. and J.Y. performed all 3D-NMR studies and determined the solution structure of Rap1b/talin-F0 complex. T.B. and S.K. performed the functional experiments showing that the double mutations of talin resulted in defective cell adhesion and spreading, and T.B., S.K. and M.M. analyzed the data. A.H. and J.H. performed the integrin activation assay in CHO A5 cells. L.Z., F.L. and J.Y. designed and performed vesicle co-sedimentation assays. L.Z. and H.L. prepared various types of cell lysates and analyzed the expression levels of desired proteins. L.Z. and K.S. prepared $^{15}$N/$^{13}$C-labeled Rap1 and $^{15}$N/$^{13}$C-labeled talin-F0 for 3D-NMR studies. T.B. and E.F.P. participated in the interpretation and preparation of the manuscript. L.Z., M.M. and J.Q. wrote the manuscript with contributions from all other co-authors.

## Additional information

**Competing interests:** The authors declare no competing financial interests.

