## [Peer Review File · Nature Communications]

Reviewers' comments:

Reviewer #1 (Remarks to the Author):

This is an interesting manuscript that attempts to answer a fundamental question: namely how are talins brought to the membrane to engage integrins and induce adhesion? The authors have concentrated on the interaction between a small G protein, Rap1, and the talin head. They have solved the structure of the complex formed between Rap1 and the F0 domain of talin and used NMR chemical shift mapping to show that the binding affinity is weak. They then generated membrane-bound Rap1 and used several methods to assess the binding of talin to Rap1 in this context. The manuscript is generally well written, with the occasional grammatical error (e.g. missing articles for some nouns) that can be corrected during the editorial process.

The major claim of the paper is that membrane-bound Rap1 binds more tightly to talin than to Rap1 that is free in solution. The evidence for this rests in part on some rather poor data. The only quantitative evidence for the binding is isothermal titration calorimetry (ITC), which showed such small heat changes that it is not clear at all whether the binding was being observed (rather than heats of dilution of the titrant, for example). The manuscript does have some promising data using sedimentation assays but as they are not truly quantitative and the protein quality was relatively poor they do not easily stand up on their own. I would suggest that the authors attempt to measure the affinities between talin-H and the two forms of Rap1 so that they can be properly compared, either by improving the ITC conditions or by using a different biophysical method. The NMR data showed that truncated Rap1 binds to talin F0 with an affinity of around 150 μ M. This should be measurable (or at least detectable) by ITC or another method such as SPR.

The paper could be improved by addressing the minor points below:

In Fig 1 the authors use chemical shift mapping to prove GTP dependency of the interaction and state (line 86) that it binds with a weaker affinity as evidenced by the smaller chemical shift changes observed. This is the case because the concentrations of the components used are below the K_d of the interaction, which must be higher than the \sim 150 μ M measured for GMPPNP-bound Rap1b. This would be worth stating in the text or legend, since smaller chemical shift changes per se do not indicate weaker binding. It is interesting to see that the GDP-bound Rap1a clearly does bind to talin-F0 to some extent. Have the authors investigated the binding of talin to GDP-bound Rap1b in the context of the membrane? This would be a more important comparison since the main thrust of the paper is that free Rap1b is not a good representation of the protein in its physiological context.

In Figure 2a chemical shift mapping with different talin constructs is presented. I appreciate that these are difficult data to deal with – the longer talin constructs are no doubt poorly behaved and the full-length protein is presumably a dimer. The annotations in this figure are however misleading. For example, the peak labelled R41 appears to be two peaks (given its width) and the red arrow suggests that it moves to the right. However, I can see two peaks, the left one has disappeared and the right one does not seem to be shifted. The same argument applies to F28, which seems to be a cluster of three or four different peaks. These two residues are within the regions of Rap1b that are involved in direct binding to talin, which may explain their choice in the figure. It would however be easier to interpret if the authors could choose more isolated peaks that have shifted and whose movements are not confounded by overlap. Such peaks were chosen to good effect in Figure 1b, which shows the chemical shift changes unequivocally.

The structural part of the paper is well presented for the most part. Can the authors please name the experiments shown in Supplementary Figure 2? I could only deduce them by searching the methods.

The samples were identified in the figure legend, but the experiments shown were not. Also, can the authors explain what the horizontal black lines mean? It is not obvious to the average reader. In Supp table 2 and Supp Fig 3a, the strands seem to be named strains.

In line 120 the authors describe a hydrophobic core involving I36/P37 of talin and I27/V29 of Rap1b. This does not correlate with the data in Supp Table 2 where I36 of talin is not close to either of these Rap1 residues. It is difficult to see the importance of these interactions without the pdb file, but in Figure 3e Rap1 I27 looks like it is on the edge of the interface and the major interactions are between V29 (Rap1) and P37 (talin). This would be helped by another figure (supplementary) that shows this core in more detail with a space-filling representation of the side chain. They then go on to argue that I27 is replaced by a His in H-Ras and that this residue is a specificity determinant. This argument would be more compelling if it were backed up by some mutagenesis. There are several other changes between H-Ras and Rap1: some of them could have allosteric effects and others are in interacting residues. For example, K31 in Rap1 is replaced by a Glu in Ras. According to Supp Table 2, this is close to talin E34 and could presumably form a salt bridge that would be lost in the Ras complex. In the absence of mutagenesis data any discussion about specificity of Ras vs Rap1 should include all of the differences between them.

Supp table 2 is very useful for a comparison of the interactions between the various Rap1 effectors. It would be useful if the authors could provide more information in the methods/legend about how it was generated. Was the analysis performed on the family of structures, on the closest to the mean or on the lowest energy structure? The criteria for interaction was defined as being a distance cutoff of 4Å. How was this applied? As an average across all the interactions involving a particular residue or the minimum distance between residues? A quick analysis of the KRIT1 structure paper (Li et al JBC Vol 287 p22317, Figure 4) suggests that this table suggests more interactions for the Rap1-KRIT1 complex.

Supp Fig 3b – the legend does not match the figure. The figure shows that talin-H does not bind Rap1 or H-Ras, while the legend suggests that the figure shows an interaction between talin-H and Rap1. Supp Fig 3c: can the authors please mark all of the talin residues that interact with Rap1.

Figure 4b – shows that the double mutant reduces integrin activation. Since it is mentioned a few lines later (line 157) how does this compare with the effect of deletion of the F0 domain? The double mutant can clearly still support some activation – can the F0 deletion?

Line 168 refers to Wikipedia, which is not acceptable in a scientific paper. This needs to be replaced by static literature reference(s), which I found after only a few minutes of searching.

Figure 4d – shows that GST-talin-H does not pull down free Rap1b, even though it did in Figure 2b. This needs to be addressed somewhere in the manuscript because I assume that it is based on the exposure of the blots. It would also be useful to show GTP-dependence of the Rap1b-talin interaction in the context of the membrane using GST pull downs.

Figure 4e – The heat changes shown are too small to give reliable data on binding affinities. The heats of dilution of talin itself could give rise to similar heat changes. What were the parameters obtained from the fit of the ITC data shown (i.e. ΔH , and N)? I find it hard to understand why the heat changes are so small when the interface has several polar/charged groups. Did the authors try varying the temperature and the buffer components (e.g. to Tris or other buffers) to improve the data? Presumably the ITC was performed at several concentrations, since the data in Figure 4e were obtained with almost exactly 10x the K_d in the cell. What happened at higher concentrations of both components? In Supp figure 5d, it appears that Talin-H does not bind to the LUV membranes by ITC,

even though it is an electrostatic interaction and Supp 4c clearly shows that it interacts almost as well with the membrane alone as it does with Rap1b anchored to the membrane. Further optimization of the ITC of Talin-H and membrane vesicles would therefore be useful.

Reviewer #2 (Remarks to the Author):

The manuscript describes the NMR structure of the talin F0 - Rap1B complex, the effect of Rap1B membrane localisation on the affinity of the interaction and the contribution of the talin-Rap1 interaction to integrin activation in a model cell system. The authors suggest that the direct interaction between talin F0 and Rap1 is a major factor in the regulation of integrin activity in mammalian cells.

The mechanisms underlying integrin activation have been studied intensively over many years, but are still not fully understood, and a large number of factors appear to be involved. It is well established that Rap1 plays a major role in integrin activation through recruitment of talin to the plasma membrane. According to the currently held view, talin recruitment requires RIAM that serves as an adaptor molecule that binds both talin and Rap1. Remarkably, the authors fail to quote the paper by Lagarrigue et al., (A RIAM/lamellipodin-talin-integrin complex forms the tip of sticky fingers that guide cell migration. *Nature Comms* 2015) which provides support for the role of a Rap1-RIAM/lamellipodin-talin complex in integrin activation.

However, the F0 domain of talin can also bind Rap1 directly, although weakly. This interaction was originally reported by Goult et al, *EMBO J* 2010, who showed that the talin F0 domain is structurally similar to the Ras/Rap interacting RA domain, and includes a canonical Rap1 binding site; however, the biological role of this interaction was not established. More recently, a direct interaction between the F0 domain of a Dictyostelium talin isoform (TalB) and Rap1 was demonstrated, and its role in cell-substrate and cell-cell adhesion demonstrated using a TalB F0 K16A mutation (equivalent to K15A of the manuscript) to disrupt the interaction with Rap1 (Plak et al, *BMC Cell Biol*, 2016).

Thus, whilst the structure of the complex between Rap1B and the Talin F0 domain reported here is useful, it yields no surprises. The major question regarding the talin-Rap1 interaction is not the structural basis for the interaction, but the contribution of this interaction to the regulation of integrin activity during cell adhesion and migration. Unfortunately, the authors fail to provide a convincing answer to this question, and, disappointingly, the interaction was only assessed in CHO cells expressing the platelet integrin α Ibb3 (one panel in Fig 4). While this assay is used routinely to analyse integrin activation, on its own, the assay falls a long way short of establishing the role of the Rap1B-Talin interaction in more physiological settings. As a result, the direct Rap1-talin pathway for integrin activation still remains a hypothesis, and this greatly reduces the impact of the manuscript

The authors need to give far more thought to how they can definitively address this issue. Talin has been shown to be crucial for integrin activation in platelets (Petrich et al *Blood* 2007) and for β 2-integrin activation in leukocytes (Lefort et al., *Blood* 2012) *in vivo*. Therefore the authors should explore the effects of the TalinF0 K15A,R35A mutation in mice. As far as *in vitro* assays go,, they could use talin-null cell lines in which talin has been shown to be essential for integrin activation (Theodosiou et al., *Elife* 2016) . Specifically, the authors could investigate the effect of the TalinF0 K15A,R35A mutation on talin targeting to the leading edge and its co-localisation with Rap1, and relate this effect to the morphology and dynamics of adhesion complexes, integrin activation in adhesions, as well as cell spreading and migration.

Additionally, there are some smaller points that need clarification.

1. The ITC results show unusually low enthalpy of the interaction. The majority of RA domains interact with their main target with enthalpies < 5 kCal/mol, presumably because ion pairs are formed at the interface. The reported talin F0-Rap1 interaction has enthalpy ~ -2 kCal/mol even when Rap1 is bound to the membrane, and close to 0 for free Rap1. The thermodynamics of the F0 – Rap1 interaction should be measured using the isolated F0 domain, using concentrations in the ITC cell comparable to K_d , which would provide a valuable comparison with other RA domains.

2. The Rap1 spectra of Fig2a suggest multiple forms of the protein, particularly noticeable for I27 and F28. If the small unlabelled signals correspond to other residues they need to be labelled appropriately. If multiple Rap1 forms were present this has to be described and discussed, as it would have a major implication for structure determination.

3. Figures 1a and b show the spectra of 1:2.5 excess of Rap1 with strong signal broadening. What is the reason for choosing this excess as the broadening is not discussed, and the text refers to the shift changes. A final point of titration with 1:10 excess would be a better illustration of the induced shifts.

4. The authors need to demonstrate that the full-length talin is in the inactive state.

5. The authors suggest that “that Rap1 not only promotes the membrane targeting but also activation of talin” (line 194) on the basis of the pull-down results of fig4f. This implies that Rap1 has a mechanism of activating talin that is not related to the membrane association. The data, however, do not support this view. Rap1 and membrane act synergistically, making the interaction significantly stronger to the Rap1/membrane system than to separate components. Enhancing the interaction with membrane through Rap1 would facilitate the displacement of the R9 from the F3 domain, like any other factor that would enhance the interaction of the talin head with membranes. That, there is no need to suggest any separate “activation” role of Rap1 based on the presented data.

6. Interpretation of Fig 4f as potent interaction of talin with Rap1 on the membrane seems an over-statement. The bands are rather weak, and the impurity bands appear to be disproportionately enhanced by the presence of Rap1. A more cautious interpretation would be appropriate.

Reviewer #3 (Remarks to the Author):

This paper is excellent. The information reported here is important and the experiments that support the outcome are well done: the structural data look good, all binding interfaces were verified by mutagenesis and compared to structurally very similar protein domains, explaining the different binding behavior. Also, the interaction studies via NMR are sound, the shifts are clear. The pull-downs, especially with the vesicles show exactly what they are meant to show, except for Suppl. Fig. 3b, where the authors see clear binding of RIAM but only a very faint band of Rap1 binding to talinH. This is not consistent with Fig.2b, where they pull down much more Rap1 in the same setup. This should be clarified.

I find it disturbing that the authors give a Wikipedia entry as a reference (line 168)?! If there is no publication stating the same facts they are hardly credible and should not be used as basis for argumentation. In science we should not use alternative facts!

The discussion is not really to the point and should be re-written. A large portion deals with the fact that the authors feel that weak protein-protein interactions are insufficiently studied, which may be true but is not the most pressing issue to discuss. It would be better to have a word in Rap1a and whether the binding to and activation of Talin is conserved among Rap1 family members.

Reviewer #1:

Comment 1. This is an interesting manuscript ... The manuscript is generally well written, with the occasional grammatical error (e.g. missing articles for some nouns) that can be corrected during the editorial process.

Response: Thank for the overall positive comments for our manuscript. We have gone through the text several times to correct the grammatical errors and some typos, and we hope the revised version is fine now.

Comment 2. ... The only quantitative evidence for the binding is isothermal titration calorimetry (ITC), which showed such small heat changes that it is not clear at all whether the binding was being observed (rather than heats of dilution of the titrant, for example). The manuscript does have some promising data using sedimentation assays but as they are not truly quantitative and the protein quality was relatively poor they do not easily stand up on their own. I would suggest that the authors attempt to measure the affinities between talin-H and the two forms of Rap1 so that they can be properly compared, either by improving the ITC conditions or by using a different biophysical method.

Response: This is an important point. In the following, we address it in two different angles.

First, we emphasize that our major goal is to show that Rap1 can have much enhanced interaction with talin upon anchoring to membrane to activate integrin. In this regard, our pull-down assay showed clear dramatic enhancement of talin binding to Rap1 upon anchoring of Rap1 to membrane (**Fig. 6c**). Our sedimentation data (**Fig. 6d**) also provided strong evidence that Rap1 enhanced talin binding to membrane. These experiments were done using high quality talin-H and Rap1 proteins with purity of above 90%. Our full-length talin was also with good purity and still in good quality after experiments (lane 1, **Fig. 6e**, and see also attached figure below) with some “impurity bands”. Those “impurity bands” are likely to be talin degradation bands due to partial cleavage of talin rod which bear exposed talin-H and show better binding capacity to membrane vesicle with or without anchored Rap1b. They are likely concentrated in the membrane pellet during spin-down experiment, which explains why the intensities are disproportional to purified full length talin in control lane. To verify this, we performed assay again and obtained the same result (**Supplementary Fig. 8b**). Nevertheless, although these pull-down and sedimentation data are qualitative, they are reproducible and convincing to support the mechanism of Rap1-mediated talin recruitment to membrane, which is further functionally supported by structure-based mutagenesis data on integrin activation (Fig. 5b) and new cell adhesion/spreading data using talin-null cells (Fig. 5c and 5d).

Second, to obtain quantitative insight into the membrane-enhanced Rap1/talin interaction, we went further to measure the K_d values for this interaction with and without anchoring Rap1 to membrane. We wish to point out that to our knowledge, it is difficult to use only one single method to quantitatively monitor protein-protein interactions with widely different affinities. This is due to the intrinsic technical limitations of various techniques such as ITC, SPR, NMR, etc. For example, NMR is a very effective method in measuring both weak and strong affinity if the measured protein complex is small in size (**ref 32**) but becomes ineffective when the protein complex size is large. Indeed, while NMR was effective to measure the weak affinity of small Rap1 bound to talin-F0 (**Fig. 2c**), it was unfeasible to measure the affinity of the large membrane-anchored Rap1 to talin-H due to the issue of severe line-broadening. Despite these limitations, the affinities, if measurable by different techniques, are comparable and should still help understand our proposed mechanism. Among four different methods (NMR, ITC, Nanotemper, SPR) we have tried, NMR (**Fig. 2c**) and Nanotemper (**Supplementary Fig.4e, new**) were effective to measure the weak K_d for talin binding to Rap1 unanchored to membrane, which gave comparable affinities ($K_d \sim 150\text{-}240 \mu\text{M}$). Such K_d was apparently too weak to be measured by ITC (**Supplementary Fig. 9c**). By contrast, only ITC, but not NMR, Nanotemper, and SPR (described below), was effective to measure the K_d of the talin interaction with Rap1 anchored to membrane (**Supplementary Fig. 7c**), which allowed us to compare the Rap1/talin affinity change with/without membrane. We fully agree with the reviewer that the ITC-based enthalpy change of talin-H/membrane-anchored Rap1 in **Supplementary Fig.7c** was modest, but we did observe consistent heat change compared to control (talin-H/membrane only) and more importantly the saturation step (in order to properly fit the curve to obtain the reliable K_d value). The change was apparently not due to heats of dilution of the titrant since the heat change was consistently bigger than the control group. The modest enthalpy change is probably the nature of this Rap1/talin interaction, which is not uncommon and also often observed in other systems, e.g., Figure 3D and 3E in Goult BT, *et al. J Biol Chem.* 288:8238-49, 2013), and Figure 5e and 5f (Zhang Y, *et al. Nat Commun.* 4:2608, 2013). To further confirm our finding, we repeated the experiment, which resulted in the similar value (see the 2nd experiment in **Supplementary Fig.7c**). While the titrations of talin-H to LUV showed slightly different background heats probably due to different batches of LUVs, the enthalpy changes and affinities of talin-H/LUV-Rap1b were consistent after subtraction in two independent trials. Thus, we are confident about the ITC data.

We also tried SPR technique extensively using a Biacore instrument (GE healthcare) to measure the talin binding to Rap1 with and without anchoring to membrane. We first immobilized either Rap1-GTP or Rap1-GDP onto a CM5 sensor chip and flow through talin-F0. The interactions underwent fast association and dissociation, indicating weak interaction (**attached Figure A and B below**) as consistent with the NMR data. However, the curves couldn't get saturation even at high concentration (highest concentration: $\sim 400 \mu\text{M}$). As a result, the affinities calculated by the software ($21.2 \mu\text{M}$ and $81.9 \mu\text{M}$ respectively) are not reliable enough. We also immobilized talin-H onto a CM5 sensor chip and flow through either Rap1-GTP or LUV-Rap1-GTP, but again we failed to see saturation of Rap1/talin-H (highest Rap1 concentration: $\sim 200 \mu\text{M}$) with the affinity fit to be around 1.61 mM (**attached Figure C below**). We did see binding curves with slow dissociation between talin-H and LUV-Rap1-GTP in the same buffer condition which indicates stronger interaction (**attached Figure D below**), however, LUV-Rap1-GTP also interacts with empty chip nonspecifically at low concentration (negative response in the curve) and resulted in awkward curve at high concentration, so we could not get a reliable fitting based on the limited data point. Although we could get an affinity of $2.18 \mu\text{M}$ based on the data points of three highest concentrations (**attached Figure D below**) which seems to be consistent with the membrane-enhanced Rap1/talin interaction, we don't think it is convincing enough to include this data. We also tried to immobilize LUV or LUV-Rap1-GTP onto a L1 sensor chip and flow through talin-H. However, we observed severe non-specific binding between talin-H and the empty chip which is filled with lipophilic modifications possibly because talin-H tends to interact with lipid molecules naturally. Thus, SPR was not effective in this particular type of binding analysis.

In conclusion, both pull-down and sedimentation methods were effective to provide systematic comparison about the binding of talin to Rap1 with and without attaching to membrane. NMR and Nanotemper were effective to measure the weak binding affinity for talin binding to Rap1 without membrane whereas only ITC was effective to measure the affinity of talin binding to Rap1 anchored to membrane. Overall, we feel the combined data by multiple techniques provide convincing evidence that upon anchoring to membrane, Rap1 binds much more strongly to talin.

The paper could be improved by addressing the minor points below:

Comment 3. In Fig 1 the authors use chemical shift mapping to prove GTP dependency of the interaction and state (line 86) that it binds with a weaker affinity as evidenced by the smaller chemical shift changes observed. This is the case because the concentrations of the components used are below the K_d of the interaction, which must be higher than the ~ 150 μM measured for GMPPNP-bound Rap1b. This would be worth stating in the text or legend, since smaller chemical shift changes per se do not indicate weaker binding. It is interesting to see that the GDP-bound Rap1a clearly does bind to talin-F0 to some extent. Have the authors investigated the binding of talin to GDP-bound Rap1b in the context of the membrane? This would be a more important comparison since the main thrust of the paper is that free Rap1b is not a good representation of the protein in its physiological context.

Response: We carefully rewrote the legend of **Fig. 2b (line 640-644)**. We agree that if smaller chemical shift changes are induced by different target proteins, they may not necessarily indicate the same extent of binding. However, both Rap1-GDP and Rap1-GMPPNP induced the same pattern of chemical shift changes of the same target ^{15}N -talin-F0, so the extent of changes should be a good indication of binding strength at the same condition. This method has been used extensively in NMR, for example, to study the mutation-induced disruption/reduction of the binding. Furthermore, our analysis is also consistent with the previously reported affinity of Rap1-GDP/talin-F0 (~ 700 μM by NMR, also see **ref 26**), which is much weaker than that of Rap1-GMPPNP/talin-F0 ($\sim 150\mu\text{M}$ in **Fig. 2c**).

Note that although the sample concentration we used for the spectral comparison was below K_d , the spectral changes were clear enough to tell the difference. This is because weak interactions in our study undergo fast exchange in NMR time scale and the peaks observed are always the average of free and bound forms, i.e., only one set of peaks, which allow us to monitor the change at such concentration (below K_d) to avoid possible precipitation etc. To further support our conclusion, we performed additional talin binding experiments involving membrane-bound Rap1-GTP and Rap1-GDP as suggested by the reviewer. These experiments utilized pull-down and sedimentation assays, which further proved the GTP dependence of the interaction (**Supplementary Fig. 7b, Supplementary Fig. 8a and 8b**).

Comment 4. In Figure 2a chemical shift mapping with different talin constructs is presented. I appreciate that these are difficult data to deal with – the longer talin constructs are no doubt poorly behaved and the full-length protein is presumably a dimer. The annotations in this figure are however misleading. For example, the peak labelled R41 appears to be two peaks (given its width) and the red arrow suggests that it moves to the right. However, I can see two peaks, the left one has disappeared and the right one does not seem to be shifted. The same argument applies to F28, which seems to be a cluster of three or four different peaks. These two residues are within the regions of Rap1b that are involved in direct binding to talin, which may explain their choice in the figure. It would however be easier to interpret if the authors could choose more isolated peaks that have shifted and whose movements are not confounded by overlap. Such peaks were chosen to good effect in Figure 1b, which shows the chemical shift changes unequivocally.

Response: We appreciate the reviewer’s understanding about the difficulty of the studies. Indeed, the longer the talin construct, the more line broadenings we saw in the NMR spectra but the line-broadening is actually another sign of binding. We apologize for the confusing labeling and have re-made the figures and re-labeled the peaks to make them clear (**Fig.3a, Supplementary Fig. 2c and 6a**). For R41 region (**Supplementary Fig. 2c**), R41 was unambiguously assigned to the right peak and clearly got shifted and broadened upon addition of different talin constructs indicating the binding. The left peak was not assigned but it is likely to be the duplex of R41 due to local conformational exchange and merged into the right peak upon binding talin. For F28 region, we unambiguously assigned F28 and V12. While it is possible that there are some weak peaks underneath F28 and V12, our bond-correlation spectra did not reveal that. To make our study more informative, we have included the profiles of chemical shift changes for readers’ convenience (**Supplementary Fig. 1a, new**). The profiles are actually fully consistent with the interface of our NMR-derived complex.

Comment 5. The structural part of the paper is well presented for the most part. Can the authors please name the experiments shown in Supplementary Figure 2? I could only deduce them by searching the methods. The samples were identified in the figure legend, but the experiments shown were not. Also, can the authors explain what the horizontal black lines mean? It is not obvious to the average reader. In Supp table 2 and Supp Fig 3a, the strands seem to be named strains.

Response: We originally mentioned the name of these two experiments in the method section, and now we also added the experiment names into the legend (please see **Supplementary Fig. 3**). The horizontal black line of each strip is diagonal line whose position indicates the chemical shift of the specific proton labeled on top of each strip, and we added this description into the legend as well. We also corrected all the “strain” into “strand”.

Comment 6. In line 120 the authors describe a hydrophobic core involving I36/P37 of talin and

I27/V29 of Rap1b. This does not correlate with the data in Supp Table 2 where I36 of talin is not close to either of these Rap1 residues. It is difficult to see the importance of these interactions without the pdb file, but in Figure 3e Rap1 I27 looks like it is on the edge of the interface and the major interactions are between V29 (Rap1) and P37 (talin). This would be helped by another figure (supplementary) that shows this core in more detail with a space-filling representation of the side chain. They then go on to argue that I27 is replaced by a His in H-Ras and that this residue is a specificity determinant. This argument would be more compelling if it were backed up by some mutagenesis. There are several other changes between H-Ras and Rap1: some of them could have allosteric effects and others are in interacting residues. For example, K31 in Rap1 is replaced by a Glu in Ras. According to Supp Table 2, this is close to talin E34 and could presumably form a salt bridge that would be lost in the Ras complex. In the absence of mutagenesis data any discussion about specificity of Ras vs Rap1 should include all of the differences between them.

Response: Thanks for very careful reading. In our structure, talin-F0 I36/P37 does form the hydrophobic core with V21/I27/V29 of Rap1b but our description and figure were not clear enough. To make it clearer, we revised our text (**line 123**) and remade **Fig. 4e** and added a new figure regarding this distinct hydrophobic core (**Supplementary Fig. 4b**). The closest distance between talin-F0 I36 (CG2) and Rap1 I27 (CD1) in our lowest energy structure is 4.1 Å, and the closest distance between talin-F0 I36 (CD1) and V29 (CG2) is 5.4 Å. Therefore, I36 was not listed in the table for contacting either I27 or V29 of Rap1 since we used 4 Å as the cut-off, but we believe I36 is part of the core considering that hydrophobic interaction still occurs within 4 ~ 6Å. As the reviewer mentioned, there indeed are several residue differences between H-Ras and Rap1, which may potentially affect talin-F0 binding. I27 and K31 of Rap1b are two strongest candidates. To verify this, we experimentally checked the effect of I27H or K31E mutation of Rap1b on the talin-F0 binding (see **Supplementary Fig. 4c, new**). We did see they still interacted with talin-F0 but much less potently. We observed overall less chemical shift changes as well as less extent of peak broadening for either I27H or K31E mutant. We have revised our description in the text as well (**line 125-128**).

Comment 7: Supp table 2 is very useful for a comparison of the interactions between the various Rap1 effectors. It would be useful if the authors could provide more information in the methods/legend about how it was generated. Was the analysis performed on the family of structures, on the closest to the mean or on the lowest energy structure? The criteria for interaction was defined as being a distance cutoff of 4Å. How was this applied? As an average across all the interactions involving a particular residue or the minimum distance between residues? A quick analysis of the KRIT1 structure paper (Li et al JBC Vol 287 p22317, Figure 4) suggests that this table suggests more interactions for the Rap1-KRIT1 complex.

Response: The residues were identified by PyMOL 1.3 (Schrödinger, LLC.) where one could label all the residues of one object that are within a specific distance cutoff (e.g. 4 Å, 5 Å or 6 Å) of another object or a specific residue. The residues were then manually organized into the table. We added this description into the legend of **Supplementary table 2**. The analysis was conducted on

each specific pdb file of the matched complex as we mentioned in **Fig. 4c**. We added the PDB code information into **Supplementary table 2** to make it clearer. For our NMR structure of Rap1/talin-F0, we analyzed the one with lowest energy. With regards to a distance cutoff of 4 Å, our reason was that the H-bonds or salt bridges are typically within 2.5-3.5 Å and they are contributing mainly to the interaction. We know that some hydrophobic interaction maybe formed above 4 Å or even 5 Å, but given that the interface of each complex structure is very extensive, we tried to simplify the comparison by listing the most critical residues. However, one could easily compare those complexes with a bigger distance cutoff in PyMOL.

We compared our table with the figure (Li *et al*, JBC Vol 287 p22317) the reviewer mentioned. It turned out that our table is consistent with that. The difference is the way how people interpreted. In the figure by Li et al, the authors connected the interacting residues by lines or dotted lines (e.g. E37 of Rap1 was connected to Y431, S433, R423 of KRIT1, and D38 of Rap1 was connected to Y431, S433 and R432 of KRIT1), but we just simply listed those residues to the specific Rap1 residue in the table. The only difference is that Li et al didn't list V21 of Rap1 in the figure probably because they focused only on the β -strand and switch regions but we did. Nevertheless, the CG2 of Rap1 V21 is close to NH2 of KRIT1 R452 with a distance of 4.0 Å.

Comment 8: Supp Fig 3b – the legend does not match the figure. The figure shows that talin-H does not bind Rap1 or H-Ras, while the legend suggests that the figure shows an interaction between talin-H and Rap1. Supp Fig 3c: can the authors please mark all of the talin residues that interact with Rap1.

Response: There was actually a faint band in the first lane of our original figure (see the attached figure A below). We apologize that we did not pay much attention to that and we replaced the figure with another one with a longer exposure time to make it look clearer (see the attached figure B below or **Supplementary Fig. 4d**). To make our data more convincing, we also attach our results of the other two independent experiments of this assay (see the attached figure C below). In addition, our NMR data also proved this point (**Fig. 2a** and **Supplementary Fig.1b**).

We re-made the figure of the original Supplementary Fig.3c and marked all the residues of talin-F0 that contact Rap1 with a distance cutoff of 4 Å (See **Supplementary Fig. 5a**).

Comment 9: Figure 4b – shows that the double mutant reduces integrin activation. Since it is mentioned a few lines later (line 157) how does this compare with the effect of deletion of the F0 domain? The double mutant can clearly still support some activation – can the F0 deletion?

Response: Talin-H_DM still supports some integrin activation probably because the F1 loop, F2 and F3 domains have membrane association ability (Goult BT *et al*, 2010 –**ref 26**. Song X *et al*, 2012- **ref 8**) or due to talin-F3/RIAM interaction (Yang J *et al*, 2014—**ref 12**), which may also contribute to the recruitment of talin-H in this system. The effect of F0 deletion in integrin activation has been studied in a previous paper (Bouaouina M *et al*, 2008, discussed in our text, **line 160-162, ref 37**) where the authors showed that F0 deletion significantly impaired the $\beta 1$ integrin activation and substantially decreased $\beta 3$ integrin activation (still supported some $\beta 3$ integrin activation). Our data is consistent with theirs. However, our data now provides a definitive structural basis of the Rap1/talin-F0 interaction in integrin activation.

Comment 10: Line 168 refers to Wikipedia, which is not acceptable in a scientific paper. This needs to be replaced by static literature reference(s), which I found after only a few minutes of searching.

Response: Thanks for the reminder. We have now cited a scientific literature (**ref 39**: Pollard TD *et al*. Molecular mechanisms controlling actin filament dynamics in nonmuscle cells. *Annu Rev*

Biophys Biomol Struct. 2000; 29:545-76.). As listed in the table 1 of this paper, an inactivated platelet contains 220 μM unpolymerized actin and 330 μM polymerized actin.

Comment 11: Figure 4d – shows that GST-talin-H does not pull down free Rap1b, even though it did in Figure 2b. This needs to be addressed somewhere in the manuscript because I assume that it is based on the exposure of the blots. It would also be useful to show GTP-dependence of the Rap1b-talin interaction in the context of the membrane using GST pull downs.

Response: We apologize for the confusion. We have revised the legend to make this clearer to readers (**line 681-686**). Yes, in previous Fig. 2b (now **Fig. 3b**), we had to load large amount of free Rap1 in order to see the very weak interaction. However, in the upper western blot panel of Fig.4d (now **Fig. 6c**), we had to keep free Rap1 low in the same amount as membrane-anchored Rap1 to avoid hugely exposed band of the latter that binds very strongly to talin, which could even be clearly visible in coomassie blue-staining gel (we boxed the band in red in revised **Fig.6c**)

Thanks for the reviewer's great suggest to test the GTP-dependence of Rap1b-talin interaction in the context of the membrane. We performed the experiments and please see our new figures **Supplementary Fig. 7b, Supplementary Fig. 8a and 8b**.

Comment 12: Figure 4e – The heat changes shown are too small to give reliable data on binding affinities. The heats of dilution of talin itself could give rise to similar heat changes. What were the parameters obtained from the fit of the ITC data shown (i.e. ΔH , and N)? I find it hard to understand why the heat changes are so small when the interface has several polar/charged groups. Did the authors try varying the temperature and the buffer components (e.g. to Tris or other buffers) to improve the data? Presumably the ITC was performed at several concentrations, since the data in Figure 4e were obtained with almost exactly 10x the K_d in the cell. What happened at higher concentrations of both components? In Supp figure 5d, it appears that Talin-H does not bind to the LUV membranes by ITC, even though it is an electrostatic interaction and Supp 4c clearly shows that it interacts almost as well with the membrane alone as it does with Rap1b anchored to the membrane. Further optimization of the ITC of Talin-H and membrane vesicles would therefore be useful.

Response: See our general response to Comment 2. To be more detailed, we provided the parameters in the figure in this revised manuscript (**Supplementary Fig. 7c**). In our original Fig.4e data (now **Supplementary Fig. 7c**, 1st batch of LUVs), the ITC data of talin-H to LUV-only displayed a similar bi-phasic heat change (heat release followed by heat absorption) for each titration, the net heat change is closed to 0. However, the titration of talin-H to LUV-Rap1 showed negative change at the beginning and then gradually reached to the background level (saturation step). The observed heat change is contributed by the type of interaction but also the number of complex formed (we have concentration limits in this study as discussed below). We used phosphate buffer because we had to use low pH ($\text{pH} < 7.0$) buffer for the membrane anchoring

experiment, and plus phosphate buffer is ideal for ITC experiments. We don't think it is a good idea to use Tris buffer for ITC experiments since Tris buffer has high heats of ionization which may result in bigger noise. We did use Hepes buffer (pH7.5) to measure the affinity of Rap1/talin-F0 by ITC but failed.

There is concentration limit for talin-H. We used 200 μ M talin-H (equivalent to \sim 10 mg/ml) since talin-H easily aggregates at room temperature at higher concentrations (e.g. 300 μ M). In order to reach a good saturation state for ITC, we had to minimize the concentration of Mem-Rap1 to 10~20 μ M. For membrane co-sedimentation in original Supplementary Fig. 4c (new **Fig 6d**), we used MLV rather than LUV (the latter is small and suitable for ITC as we mentioned in the method section). MLV is larger than LUV and could be pelleted down easily at normal high speed. Furthermore, in this assay, MLV contained 10% PIP2, which was reported to be favorable for binding to talin-H (**ref 8**) and MLV-10%PIP2 was also a homogenous solution that was found to be suitable for Rap1 anchoring but not MLV-1%PIP2. Under these conditions, it is not surprising to see talin-H binding to the MLV membrane. In any case, our goal here in this sedimentation assay was to show that membrane-anchored Rap1 has enhanced interaction with talin-H or full length talin than membrane only.

Reviewer #2 (Remarks to the Author):

Comment 1:... According to the currently held view, talin recruitment requires RIAM that serves as an adaptor molecule that binds both talin and Rap1. Remarkably, the authors fail to quote the paper by Lagarrigue et al., (A RIAM/lamellipodin–talin–integrin complex forms the tip of sticky fingers that guide cell migration. Nature Comms 2015) which provides support for the role of a Rap1-RIAM/lamellipodin-talin complex in integrin activation.

Response: RIAM is likely an important molecule. However, as we explained in the introduction, our rationale for the study was to understand why mice without RIAM are viable and functions of many cells including platelets are normal. Given that RIAM/talin may be one of the pathways to recruit talin for activating integrin, we wondered whether direct Rap1/talin interaction could be another pathway. Our results including all newly generated functional data (as requested by the reviewer, see below) strongly demonstrate that this pathway is possible. We did not discuss Lagarrigue et al., Nat Commun 2015 paper that focused on elucidating how RIAM may mediate cell migration since our manuscript focused on how Rap1/talin interaction regulates integrin activation – the initial step of the cell adhesion. However, we did propose in the discussion of the original manuscript that talin/RIAM pathway could complement with talin/Rap1 pathway for activating integrin (**line 254-257**). In the revised manuscript, we revised the discussion suggesting that specific RIAM containing MIT complex proposed by Lagarrigue et al (now **ref 44**) may represent the alternative talin/RIAM pathway for triggering integrin activation (**line 252-254**).

Indeed, this pathway may explain why Rap1 binding defective talin_DM still partially supports integrin activation in our systems (see **line 248-257**).

Comment 2: ... a direct interaction between the F0 domain of a Dictyostelium talin isoform (TalB) and Rap1 was demonstrated, and its role in cell-substrate and cell-cell adhesion demonstrated using a TalB F0 K16A mutation (equivalent to K15A of the manuscript) to disrupt the interaction with Rap1 (Plak et al, BMC Cell Biol, 2016). Thus, whilst the structure of the complex between Rap1B and the Talin F0 domain reported here is useful, it yields no surprises. The major question regarding the talin-Rap1 interaction is not the structural basis for the interaction, but the contribution of this interaction to the regulation of integrin activity during cell adhesion and migration. Unfortunately, the authors fail to provide a convincing answer to this question, and, disappointingly, the interaction was only assessed in CHO cells expressing the platelet integrin α IIb β 3 (one panel in Fig 4). While this assay is used routinely to analyse integrin activation, on its own, the assay falls a long way short of establishing the role of the Rap1B-Talin interaction in more physiological settings. As a result, the direct Rap1-talin pathway for integrin activation still remains a hypothesis, and this greatly reduces the impact of the manuscript. The authors need to give far more thought to how they can definitively address this issue. Talin has been shown to be crucial for integrin activation in platelets (Petrich et al Blood 2007) and for β 2-integrin activation in leukocytes (Lefort et al., Blood 2012) in vivo. Therefore the authors should explore the effects of the TalinF0 K15A,R35A mutation in mice. As far as in vitro assays go, they could use talin-null cell lines in which talin has been shown to be essential for integrin activation (Theodosiou et al., Elife 2016) . Specifically, the authors could investigate the effect of the TalinF0 K15A,R35A mutation on talin targeting to the leading edge and its co-localisation with Rap1, and relate this effect to the morphology and dynamics of adhesion complexes, integrin activation in adhesions, as well as cell spreading and migration.

Response: We appreciate the reviewer's critical comment and agree that the physiological relevance of human Rap1/talin interaction is clearly the key concern here. Before we address that, we wish to first emphasize that the weak human Rap1/talin interaction has been long ignored likely due to two reasons we described in the introduction (**line 67-71**) despite the reported analysis on the stronger interaction between Dictyostelium talin isoform (TalB) and Rap1 ($K_d \sim 20\mu\text{M}$). Thus solving the structure was crucial to find out how the weak interaction may occur, which guided us to select specific mutations to evaluate the importance of the interaction. In the original paper (**ref 26**) that reported human Rap1/talin interaction, a few residues were suggested to be in the interface, e.g., R30, but our structure showed this R30 residue was not in the interface at all strongly demonstrating the importance of the structural work presented in this manuscript (we selected K15/R35 for mutations based on the structure). In other words, had we made a R30A mutation and seen no functional defect in integrin activation, we would have reached totally opposite conclusion. Furthermore, our structure allowed us to develop an idea that Rap1/talin-H and membrane/talin-H interactions may have synergetic effect (**Fig.6b**) in that the C-terminus of Rap1 known to be attached to membrane via prenylation aligns well with talin-H that has the positive charged surface when they formed complex. We then performed the pull-down and ITC experiments, and obtained

exciting data that membrane-anchored Rap1 binds much stronger to talin than free Rap1. This is a significant conceptual advance further demonstrating the importance of our structural analysis.

Next, with regards to the physiological data of the Rap1/talin interaction, we agree that CHO cell data showing reduced integrin activation by talin-H_DM is just one functional evidence but as reviewer indicated, this approach has been widely used and appreciated in the field. Given that we have extensive structural, biochemical, and mutagenesis data, we believe this key functional data provides definitive evidence about the importance of human Rap1/talin interaction in regulating integrin activation. Following the reviewer's suggestion to gain further physiological evidence of the interaction, we sought for collaboration with the cell adhesion genetics expert Dr. Markus Moser in Max Planck Institute, Germany. Through major effort, Dr Moser's group succeeded in expressing WT talin and Rap1 binding defective talin_DM respectively in talin null cells and they then performed extensive cell adhesion and spreading experiments. New **Fig.5c** demonstrates that the cell adhesion to multiple integrin ligands fibronectin, vitronectin, and laminin were substantially reduced for talin_DM-expressed cells as compared to those for WT talin, which is fully consistent with the defect in integrin activation in **Fig.5b** for talin-H_DM. New **Fig.5d** further demonstrates the cell spreading by talin_DM was significantly impaired. All the protein expression controls are provided in new **Supplementary Fig. 6b**. These new functional data thus provide additional strong physiological evidence of the human Rap1/talin interaction in regulating integrin activation and cell adhesion. We agree that talin_DM knock-in mice may also provide information about the role of the Rap1/talin interaction. However, this kind of genetic work requires tremendous amount of time and effort, which is totally beyond the scope of this structure-based study. We strongly feel that our extensive functional data in **Fig 5b, c, d** already provided sufficient evidence for the importance of the Rap1/talin interaction in regulating integrin activation and cell adhesion. The combined structural, biochemical, and functional data will significantly advance our understanding of the talin-mediated integrin activation and hopefully further trigger more fertile investigations including above-mentioned genetic work.

Additionally, there are some smaller points that need clarification.

Comment 3. The ITC results show unusually low enthalpy of the interaction. The majority of RA domains interact with their main target with enthalpies < 5 kCal/mol, presumably because ion pairs are formed at the interface. The reported talin F0-Rap1 interaction has enthalpy ~ -2 kCal/mol even when Rap1 is bound to the membrane, and close to 0 for free Rap1. The thermodynamics of the F0 – Rap1 interaction should be measured using the isolated F0 domain, using concentrations in the ITC cell comparable to K_d , which would provide a valuable comparison with other RA domains.

Response: Please see the responses to reviewer 1 – comment 2 and comment 12. We did try to measure the Rap1/talin-F0 interaction by ITC at higher concentration, but we could not reach saturation apparently due to very weak interaction (see the attached figure below). Note that we used 0.7 mM talin-F0 in this experiment, but the observed heat change was still small. The enthalpy

change of Rap1/talin interaction is smaller than other Rap1/RA domain interactions, but we did observe consistent values in two independent trials (**Supplementary Fig.7c**).

In this revised manuscript, we included the quantitative comparison of RIAM and talin in terms of binding Rap1. We measured the affinities of Rap1/RIAM, Rap1/talin-F0 and Rap1/talin-H by another instrument called Nanotemper (**Supplementary Fig. 4e, new**). Rap1/talin-F0 interaction was around 230 μM which is pretty similar to our NMR titration (**Fig. 2c**). Rap1/talin-H was estimated to be around 150 μM (not saturated due to aggregation issue at high concentration). Rap1-GTP/RIAM was around 40 times stronger ($\sim 5 \mu\text{M}$) which is consistent with our GST-pull down data (**Supplementary Fig. 4d**).

It should be mentioned again (see the response to reviewer 1 comment 2) that the detection of such weak interaction is a big challenge and reaches the limit to many current biochemical techniques especially when the protein concentration itself is a limitation. For example, we had to use 1 mM talin-F0 to reach the saturation in our Nanotemper assay, however, we couldn't get saturation in the case of talin-H since 0.3 mM talin-H ($\sim 15 \text{ mg/ml}$) easily aggregated at room temperature by itself.

Comment 4. The Rap1 spectra of Fig2a suggest multiple forms of the protein, particularly noticeable for I27 and F28. If the small unlabelled signals correspond to other residues they need to be labelled appropriately. If multiple Rap1 forms were present this has to be described and discussed, as it would have a major implication for structure determination.

Response: We apologize for the confusing labeling. We re-made and re-labeled those figures (now **Fig.3a, Supplementary Fig. 2c and 6a**). We did see some duplex peaks for some residues in our spectrum (e.g. I27) likely due to the conformational flexibility of Rap1 protein especially in the switch regions but most of these peaks merged into single peaks in the complex (described in **line 371-376**). We used free form of Rap1b for assignment experiments and did see some duplex peaks

for some residues. However, we used data collected from complex form (three different 3D-NOESY experiments) to calculate structure with one major set of peaks..

Comment 4. Figures 1a and b show the spectra of 1:2.5 excess of Rap1 with strong signal broadening. What is the reason for choosing this excess as the broadening is not discussed, and the text refers to the shift changes. A final point of titration with 1:10 excess would be a better illustration of the induced shifts.

Response: Fig.2a and b (original Fig.1a and b) showed those spectra in the same contour level. We attached the profile of the chemical shift changes of ^{15}N labeled talin-F0 in this revised manuscript (**Supplementary Fig.1a, new**). We would like to mention that at this 1:2.5 ratio, only three residues were completely broadened including T16, I36, and E38 probably due to chemical exchange (mentioned in the figure legend). We chose 1:2.5 ratio just simply because of the experimental convenience (e.g. stock protein concentration). We compared Rap1-GDP, Rap1-GMPPNP, and HRas-GMPPNP in terms of binding ^{15}N -talin-F0 at the same time. We purified them and measured the concentrations at the same time to make sure the assays were conducted in the exact same conditions. We initially thought that H-Ras could interact with talin-F0 due to high sequence homology to Rap1 but it turned out that H-Ras didn't interact with talin-F0 at all demonstrating strong specificity of Rap1 binding to talin.

1:10 excess resulted in the same pattern of changes but bigger chemical shift change and more peak broadenings. We think 1:2.5 ratio is a better illustration in the sense of the consistency of our work since we have Rap1-GDP and HRas-GMPPNP as the comparison in the same ratio (**Fig.2b and supplementary Fig.1b**). In the figure below, we just want to show the reviewer the 1:10 ratio spectrum with more peaks broadened than 1:2.5 ratio.

Comment 6. The authors need to demonstrate that the full-length talin is in the inactive state.

Response: Purified full-length talin has been shown to adopt a compact and auto-inhibitory structure (Goult, BT *et al*, 2013, **ref 10**). Below we show NMR data where ^{15}N -labeled integrin $\beta 3$ cytoplasmic tail binds potently to talin-F2F3 but minimally to full length talin at the same condition, confirming the autoinhibition of talin. We do not feel it is necessary to present this in the formal manuscript since it is not a new finding (we already have 6 main figures and 9 supplementary figures). In addition, our vesicle co-sedimentation experiments also provide proof for the autoinhibition. In **Fig.6d and 6e**, full-length talin does not interact with membrane vesicle as potently as talin-H. Quantitatively, the membrane vesicle could pellet down ~20-25 % of talin-H but only around 2-3% of full-length talin apparently due to the autoinhibition state of full length talin.

Comment 7. The authors suggest that “that Rap1 not only promotes the membrane targeting but also activation of talin” (lane 194) on the basis of the pull-down results of fig4f. This implies that Rap1 has a mechanism of activating talin that is not related to the membrane association. The data, however, do not support this view. Rap1 and membrane act synergistically, making the interaction significantly stronger to the Rap1/membrane system than to separate components. Enhancing the interaction with membrane through Rap1 would facilitate the displacement of the R9 from the F3 domain, like any other factor that would enhance the interaction of the talin head with membranes. That, there is no need to suggest any separate “activation” role of Rap1 based on the presented data.

Response: What we meant was that upon talin recruitment to membrane by Rap1, the interaction of membrane with talin may further assist or promote talin unmasking via the pull-push mechanism (**ref 8**). We agree with the reviewer that Rap1 itself mainly plays the role in recruiting talin to the membrane site and the PIP2 lipids are direct talin activator. On the other hand, since Rap1 is known to be anchored at the membrane, it may also help re-orientate talin to membrane resulting in the

membrane-mediated conformational change of talin although such speculation remains to be proved. We have revised our description (**line 218-221**, and **line 241-246**).

Comment 8. Interpretation of Fig 4f as potent interaction of talin with Rap1 on the membrane seems an over-statement. The bands are rather weak, and the impurity bands appear to be disproportionally enhanced by the presence of Rap1. A more cautious interpretation would be appropriate.

Response: Thanks for these comments. Please see our response to reviewer 2's comment 2. Again, we believe that those "impurity bands" are partial cleavage of talin rod but they are less than 10% in our talin input (lane 1, **Fig. 6e**). They showed up disproportionally in the presence of anchored Rap1 because these degraded talins are likely to adopt an exposed talin-H and show better binding capacity to membrane vesicle with or without anchored Rap1b and also get concentrated on membrane pellets. We repeated the experiment (**supplementary Fig. 8b**) and obtained the same observation. We also revised our description (**line 213-216**).

Reviewer #3 (Remarks to the Author):

Comment 1: This paper is excellent. The information reported here is important and the experiments that support the outcome are well done: the structural data look good, all binding interfaces were verified by mutagenesis and compared to structurally very similar protein domains, explaining the different binding behavior. Also, the interaction studies via NMR are sound, the shifts are clear.

Response: Thanks for the positive comment on our work.

Comment 2: The pull-downs, especially with the vesicles show exactly what they are meant to show, except for Suppl. Fig. 3b, where the authors see clear binding of RIAM but only a very faint band of Rap1 binding to talinH. This is not consistent with Fig.2b, where they pull down much more Rap1 in the same setup. This should be clarified.

Response: Yes, the band was actually faint with consistency. We replaced the original figure (attached figure A below) with another one with a longer exposure time to make it look better (see the attached figure B below or our new **Supplementary Fig. 4d**). To make our data convinced, we also attached our results of the other two independent experiments of this assay (see the attached figure C below). Indeed, the band of Rap1 pulled down by GST-talin-H in this experiment looks fainter compared to that in **Fig. 3b** (original Fig. 2b) although they were done in the same condition. The reason is that we used anti-His tag antibody to detect the band of either Rap1 or H-Ras (they are both His-tagged) in this experiment, but we used anti-Rap1 antibody in **Fig. 3b** (original Fig. 2b). According to our experience, the sensitivity of anti-Rap1 antibody seems to be

better than that of anti-His tag antibody in terms of recognizing His-tagged Rap1. However, anti-His tag antibody is apparently more suitable for this experiment to prove our point that GST-RIAM binds equally well to both Rap1 and H-Ras but not for GST-talin-H.

Comment 3: I find it disturbing that the authors give a Wikipedia entry as a reference (line 168)?! If there is no publication stating the same facts they are hardly credible and should not be used as basis for argumentation. In science we should not use alternative facts!

Response: Thanks for the comment. A citation (now **ref 39**) has been added (Pollard TD *et al.* Molecular mechanisms controlling actin filament dynamics in nonmuscle cells. *Annu Rev Biophys Biomol Struct.* 2000; 29:545-76.). As listed in the table 1 of this paper, an inactivated platelet contains 220 μ M unpolymerized actin and 330 μ M polymerized actin.

Comment 4: The discussion is not really to the point and should be re-written. A large portion deals with the fact that the authors feel that weak protein-protein interactions are insufficiently studied, which may be true but is not the most pressing issue to discuss. It would be better to have a word in Rap1a and whether the binding to and activation of Talin is conserved among Rap1 family members.

Response: Thanks for your suggestion. Rap1a and Rap1b are highly conserved with overall ~95 % identity and fully conserved in the binding interface (**Supplementary Fig. 5b**). We have revised our discussion (**line 145-146**, and **line 230-231**).

Reviewers' comments:

Reviewer #1 (Remarks to the Author):

The manuscript is much improved.

The extra experimental data that the authors have provided, along with their extensive revision of the manuscript, has strengthened the manuscript considerably. The majority of their claims are now well supported and those that are weaker are discussed appropriately.

All of the issues and points that I raised in my original review have now been addressed and I do not have any more queries about the biophysical and structural parts of the work described.

Reviewer #2 (Remarks to the Author):

The authors addressed many technical points and, more importantly, introduced new biological experiments to support the contribution of the talin-Rap1 interactions into the regulation of integrin activity. They also improved the descriptions in the text and made discussion more rounded.

However, there are still several major shortcomings that make the manuscript not fully convincing.

1. New cell data are incomplete and unreliable without analysis of cell morphology and talin distribution throughout the cell. Cell images showing normal cell spreading and adhesion formation after transformation need to be presented as a quality control. If DM-mutant has compromised targeting to the membrane and adhesion complexes, as the paper suggests, one would expect to detect clear decrease of the talin level at the plasma membrane and adhesions, compared to the WT. Possibly, adhesion complexes would be smaller in or their number decrease. Since the authors used fluorescent talin in transfections, these images would be very straightforward to obtain and analyse. I have suggested these experiments in previous review. "Specifically, the authors could investigate the effect of the TalinF0 K15A,R35A mutation on talin targeting to the leading edge and its co-localisation with Rap1, and relate this effect to the morphology and dynamics of adhesion complexes, integrin activation in adhesions, as well as cell spreading and migration."
2. Integrin activation data of fig.5b lack evidence of the same expression level of WT and DM.
3. Intensities of the positive bands of the pull-down experiments of fig.6c and e are extremely low, comparable or even less than some of the impurities. This makes conclusions from the experiments rather tentative and does not fully justify statements in the text that "membrane-anchored Rap1 binds more robustly... to talin H" or "more potently" to FL talin. The increase of talin bound to the membrane in the presence of Rap1 (fig.6d) is rather modest, not as dramatic as 100-fold increase in the binding affinity suggested by ITC. It is far from clear how these binding characteristics would translate into cell environment. Data on co-localisation of talin-WT and DM with Rap1 and localisation on the membrane in cells are critical, before a reliable conclusion can be reached.
4. The only evidence for the 100-fold increase in affinity through the synergy between membrane and Rap1 interactions is the ITC experiment. This is a very important conclusion, with an additional surprise of the low enthalpy of the interaction despite expected charge interactions of talin with both Rap1 and membrane. Due to the critical role of the result, supplementary panel fig.7a needs to be in the main figure, as in the original version. Low enthalpy of the interaction needs to be discussed and compared to the enthalpies of other Rap1 interactions. A critical control ITC experiment testing the interaction of talin with LUV-Rap1/GDP is missing.

Response to reviewers' comments

No concerns from reviewer 1 and 3.

Reviewer #2

General comment:

The authors addressed many technical points and, more importantly, introduced new biological experiments to support the contribution of the talin-Rap1 interactions into the regulation of integrin activity. They also improved the descriptions in the text and made discussion more rounded.

Response: Thanks for the positive general comment.

Specific comments:

1. New cell data are incomplete and unreliable without analysis of cell morphology and talin distribution throughout the cell. Cell images showing normal cell spreading and adhesion formation after transformation need to be presented as a quality control. If DM-mutant has compromised targeting to the membrane and adhesion complexes, as the paper suggests, one would expect to detect clear decrease of the talin level at the plasma membrane and adhesions, compared to the WT. Possibly, adhesion complexes would be smaller in or their number decrease. Since the authors used fluorescent talin in transfections, these images would be very straightforward to obtain and analyse...

Response: As requested, we now show the phase contrast pictures as well as immunofluorescence stainings in **new Supplementary Fig. 7c, d.** These pictures clearly show that upon expression of WT talin and talin DM, the talin^{1/2dko} cells are able to spread on fibronectin (FN). In both groups of cells, WT and mutant talin could localize to paxillin positive focal adhesions. We then tried to measure whether the ypet-tagged talin DM mutant could affect the size and number of focal adhesions or show decreased recruitment to focal adhesions. To this end, we seeded the cells on FN-coated micropatterns and determined the focal adhesion number and size by using paxillin as a FA marker. These experiments revealed that indeed, as speculated by the reviewer, the total adhesion area and number per cell are significantly reduced in cells expressing talin DM compared to those expressing WT talin (**new Fig. 5e-5g**). We also measured the intensity of ypet fluorescence within the focal adhesion area and normalized it to the overall cellular ypet intensity determined by epifluorescence microscopy. **Fig. 5i** shows that the relative ypet intensity in the focal adhesion area of talin DM cells is significantly reduced compared to WT talin expressing cells. We note that mean focal adhesion size was also reduced by talin DM (**Fig. 5h**), although the P Value of 0.09 is slightly larger than normally defined cut-off (0.05). We

believe all these data combined with **Fig. 5c, d** provide a convincing conclusion that impaired talin/Rap1 binding affects the recruitment of talin to FAs and integrin activation.

We wish to emphasize again as we did in the discussion that our study uncovered the role of Rap1/talin interaction as a new pathway to mediate integrin adhesion but does not exclude other possible pathways such as Rap1/RIAM. In the discussion of the re-revised manuscript, we also added that PIPKI γ and its product PIP2 may act as another pathway for promoting talin recruitment as suggested before (**line 268-269**). The existence of these pathways may explain why our double mutations did not completely abolish the integrin activation and adhesion.

2. Integrin activation data of fig.5b lack evidence of the same expression level of WT and DM.

Response: This was not requested before. We did not intend to present this kind of raw data due to space limitation but as requested now, we attach a new figure (new **Supplementary Fig. 7a**) to show the same expression level of WT and DM talin-H in transfected CHO A5 cells.

3. Intensities of the positive bands of the pull-down experiments of fig.6c and e are extremely low, comparable or even less than some of the impurities. This makes conclusions from the experiments rather tentative and does not fully justify statements in the text that “membrane-anchored Rap1 binds more robustly... to talin H” or “more potently” to FL talin. The increase of talin bound to the membrane in the presence of Rap1 (fig.6d) is rather modest, not as dramatic as 100-fold increase in the binding affinity suggested by ITC. It is far from clear how these binding characteristics would translate into cell environment. Data on co-localisation of talin-WT and DM with Rap1 and localisation on the membrane in cells are critical, before a reliable conclusion can be reached.

Response: Please note that the upper panel of **Fig. 6c** is western blot, which is sensitive and widely used, showing dramatic difference between positive band and control. To be confirmatory, the lower panel is SDS-PAGE, which shows all proteins including impurities. It is true that the positive Rap1 band in the lower panel is not strong but it shows clear difference, i.e., the control has no band! It is very common that the intensity of pulled-down prey protein band is weaker than those of GST-protein input bands in this kind of GST pull-down experiment when analyzed by SDS-PAGE since the band intensity depends on multiple factors such as the amount of input, the experimental condition as well as the affinity of the interaction. However, the key point is that the Rap1 band is not detected in the control, which is clear in the figure. The quantitative comparison was made by combined NMR and ITC measurements as we have shown in the paper, which revealed $\sim 1-3 \mu\text{M}$ for membrane-anchored Rap1/talin interaction (**Supplementary Fig. 8b**). While the affinity is not nM tight, it was truly enhanced by $\sim 50-100$ folds due to the presence of membrane.

In Fig. 6e, we agree that talin bands spun down by membrane vesicles are relatively weaker but reproducible (**Supplementary Fig. 9b**) and the issue has already been discussed in the manuscript (see **line 227-230**), “Although full-length talin binds to membrane **more weakly**

than talin-H possibly due to some degree of autoinhibition for the former”). We further point out here that despite some degree of autoinhibition, the membrane-enhanced binding is still potent, i.e., nearly 3-fold (**Fig. 6e**). Full activation of talin may be achieved in vivo by one or multiple mechanisms as described in **line 54-55** (ref 7-10), which would bind membrane-Rap1 more potently like talin-H (no autoinhibition).

In Fig. 6d, our vesicle co-sedimentation assay showed that the presence of anchored Rap1 enhanced membrane/talin-H interaction by ~1.5 folds. We note that this is a comparison between Mem-Rap1 and membrane in terms of binding talin-H. This is quite different from the “100-fold increase”, which refers to the comparison between Mem-Rap1 and free Rap1 in terms of binding talin-H. These data were obtained by completely different experiment strategies and are not quantitatively comparable, but they reached the same conclusion that “Rap1/talin” interaction synergizes with “membrane/talin” interaction.

With regards to the co-localization experiment, we performed a series of co-localization studies of talin and Rap1 on our cells by using anti-Rap1 antibodies or by transfecting mCherry-tagged Rap1 (G12V) construct in an overexpressed system. Both approaches failed either due to no specific cellular stainings or no clear membrane localization of the overexpressed mCherry-tagged Rap1. Thus we were unfortunately unable to provide direct evidence to address this point. However, our new data have clearly showed that talin-DM which does not bind Rap1 is less recruited to FAs. Despite this caveat, the extensive structural, biochemical, cell biological data presented in this revised manuscript have provided convincing evidences that Rap1/talin represents a new pathway for regulating integrin activation and adhesion.

4. The only evidence for the 100-fold increase in affinity through the synergy between membrane and Rap1 interactions is the ITC experiment. This is a very important conclusion, with an additional surprise of the low enthalpy of the interaction despite expected charge interactions of talin with both Rap1 and membrane. Due to the critical role of the result, supplementary panel fig.7a needs to be in the main figure, as in the original version. Low enthalpy of the interaction needs to be discussed and compared to the enthalpies of other Rap1 interactions. A critical control ITC experiment testing the interaction of talin with LUV-Rap1/GDP is missing.

Response: As we already pointed out in previous reply, in addition to ITC data, our GST-pull down assay (**Fig. 6c**) **also showed a dramatic enhancement** of talin/Rap1 interaction when Rap1 is anchored to membrane. ITC provided quantitative increase but the “**dramatic enhancement**” conclusion can be drawn from both pull-down and ITC data. We placed the pull-down data in the main text but placed the ITC data in the supplementary figure simply because of the figure space and organization conveniences.

We are unable to provide comprehensive enthalpy change comparison between membrane Rap1/talin and other Rap1/effector interactions due to little information on those reported

complexes. The affinity of Rap1-KRIT1 is $\sim 0.36 \mu\text{M}$ and the exact enthalpy-change value was not provided (Gingras A R *et al.* J Biol Chem. 2013). We also could not find any enthalpy information for Rap1/RIAM or Rap1/cRaf1 in literatures (mentioned in **Fig. 4c**). Rap1/SPN-ARR interaction (affinity $\sim 0.2 \mu\text{M}$) is the only one that has reported enthalpy change value: $\sim 5300 \text{ cal/mol}$ (**ref 43**), which is ~ 2 -fold bigger than that of Mem-Rap1/talin-H. We are not certain if the two fold enthalpy difference can be correlated with the affinity difference between Mem-Rap1/talin-H and Rap1/SPN-ARR, i.e., $1.5 \mu\text{M}$ for the former and $0.2 \mu\text{M}$ for the latter. Nevertheless, even if the enthalpy information for all these complexes were available, we do not feel that their comparison would provide any correlation with the specificity since there are substantial differences in primary sequences (**Fig. 4d and Supplementary table 2**) between talin-F0 and other Rap1 effectors. Rather, we believe that our comparison of primary sequence difference (**Fig. 4d and Supplementary table 2**) vs interface difference (**Fig. 4b, 4c and Supplementary table 2**) provided comprehensive insights into the specificity, which is the key body of the paper.

With regards to requested ITC data on talin/LUV-Rap1/GDP binding, we wish to note that this was not requested in the first round of review. Moreover, we must point out that we already showed the GTP dependency of talin/Rap1 interaction in multiple assays (**Fig.2b, 3b, Supplementary Fig. 8c, 9a and 9b**). After establishing this GTP dependence, our goal was then to show that membrane dramatically enhances the activated GTP-Rap1 binding to talin, which is the key achievement of the paper. In this regard, we do not feel necessary to pursue the affinity of talin/LUV-Rap1-GDP. Furthermore, based on the data in **Fig.2b, 3b, Supplementary Fig. 8c, 9a and 9b**, LUV-Rap1-GDP/talin interaction would be clearly much weaker than LUV-Rap1-GTP/talin and thus technically difficult to be measured by ITC reliably (note that ITC is mostly effective to detect strong binding with the affinity $<10 \mu\text{M}$).

REVIEWERS' COMMENTS:

Reviewer #2 (Remarks to the Author):

The authors successfully addressed my comments and included comprehensive illustrations and descriptions of the supporting data. The descriptions and discussions are now much more rounded and objective. I don't have any further comments. The study will be of high interest to researches in the adhesion field.